# Neural mechanism of acute stress regulation by trace aminergic signalling in the lateral habenula in male mice

Soo Hyun Yang [1,6], Esther Yang[1,6], Jaekwang Lee[2,6], Jin Yong Kim[1,6], Hyeijung Yoo[1], Hyung Sun Park[1], Jin Taek Jung [1], Dongmin Lee [1], Sungkun Chun [3], Yong Sang Jo [4], Gyeong Hee Pyeon[4], Jae-Yong Park [5], Hyun Woo Lee[1] ✉ & Hyun Kim [1] ✉

Stress management is necessary for vertebrate survival. Chronic stress drives depression by excitation of the lateral habenula (LHb), which silences dopaminergic neurons in the ventral tegmental area (VTA) via GABAergic neuronal projection from the rostromedial tegmental nucleus (RMTg). However, the effect of acute stress on this LHb-RMTg-VTA pathway is not clearly understood. Here, we used fluorescent in situ hybridisation and in vivo electrophysiology in mice to show that LHb aromatic L-amino acid decarboxylase-expressing neurons (D-neurons) are activated by acute stressors and suppress RMTg GABAergic neurons via trace aminergic signalling, thus activating VTA dopaminergic neurons. We show that the LHb regulates RMTg GABAergic neurons biphasically under acute stress. This study, carried out on male mice, has elucidated a molecular mechanism in the efferent LHb-RMTg-VTA pathway whereby trace aminergic signalling enables the brain to manage acute stress by preventing the hypoactivity of VTA dopaminergic neurons.

The lateral habenula (LHb) is a brain region important for the response to depression, stress, pain, anxiety, fear, aversive motivation and reward[1–9]. Human and animal studies suggest that the LHb is a powerful negative regulator of dopaminergic systems in the midbrain[10]. The dopaminergic pathway that projects from the ventral tegmental area (VTA) to the nucleus accumbens (NAc) is believed to be a key pathogenic component associated with chronic stress-induced depression[10, 11].

Unlike chronic stress, acute stress usually does not lead to depression, presumably owing to protective brain mechanisms that operate in acute stress conditions. The current understanding of the neural circuitry between the LHb and the rostromedial tegmental nucleus (RMTg) is that increased activity of LHb glutamatergic neurons may increase RMTg GABAergic neuronal activation, leading to hypoactivity of VTA dopaminergic neurons[12–14]. Paradoxically, however, various types of acute stress are known to excite both LHb glutamatergic neurons and VTA dopaminergic neurons[15–18]. As RMTg GABAergic neurons receive a large amount of input from the LHb[19], acute stress-induced activation of the LHb and the VTA is not consistent with the known function of the LHb-RMTg-VTA pathway. Therefore, it is important to determine whether the efferent pathway from the LHb to the RMTg could play an inhibitory role in the circumstance of acute stress.

The excitatory neurons in the LHb have long been considered the brain's 'anti-reward centre', upregulating the activity of the RMTg[20]. However, previous studies have confirmed the existence of

[1]Department of Anatomy, College of Medicine, Korea University, Seoul 02841, South Korea. [2]Division of Functional Food Research, Korea Food Research Institute, Wanju 55365, South Korea. [3]Department of Physiology, Jeonbuk National University Medical School, Jeonju 54907, South Korea. [4]School of Psychology, Korea University, Seoul 02841, South Korea. [5]Department of Integrated Biomedical and Life Science, Graduate School, Korea University, Seoul 02841, South Korea. [6]These authors contributed equally: Soo Hyun Yang, Esther Yang, Jaekwang Lee, Jin Yong Kim. ✉e-mail: biocais@korea.ac.kr; kimhyun@korea.ac.kr

'D-neurons' that express aromatic L-amino acid decarboxylase (AADC; also known as dopa decarboxylase or DDC) and produce trace amines, such as β-phenylethylamine, p-tyramine and tryptamine, rather than monoamines, such as dopamine and serotonin, in the LHb[21,22]. These trace amines are biosynthesised by AADC from L-phenylalanine, L-tyrosine or L-tryptophan, respectively[23], and their binding to the trace amine-associated receptor 1 (TAAR1) plays essential roles in neurotransmission and neuromodulation[24]. Notably, TAAR1-mediated signalling inhibits the firing frequency of monoaminergic neurons in the midbrain[25,26]. Changes in trace amine levels, which are normally present at very low concentrations of less than 10 nM in the brain and are not stored in vesicles, are closely related to psychiatric disorders such as depression and schizophrenia[23,27–31]. Although D-neurons exist in the LHb, studies have not yet investigated, to our knowledge, whether stress affects D-neurons in the LHb-RMTg pathway to regulate the development of depression.

In this study, we anatomically and functionally reveal the neural circuitry of LHb D-neurons for stress management. We show that, under acute stress, LHb D-neurons biphasically regulate the RMTg through trace amines, preventing a decrease in the activity of VTA dopaminergic neurons. In rodents, the reduction of trace amines in the LHb led to depressive-like behaviours such as anhedonia and despair. These behaviours were rescued by restoring trace aminergic signalling in the LHb-RMTg pathway.

Most humans do not develop depression in the face of acute stressors because of the potential involvement of protective molecular or neural mechanisms. Hence, we propose that trace aminergic signalling in the LHb-RMTg-VTA neural circuit is a candidate for that protective molecular mechanism.

## Results

### Acute stress activates AADC-expressing D-neurons in the LHb
D-neurons are located in various parts of the brain, including the LHb (Supplementary Fig. 1a). To clarify the function of the LHb-RMTg-VTA pathway in response to acute stress (Fig. 1a) and to gain molecular insights into the function of D-neurons in the LHb, we first performed fluorescence in situ hybridisation (FISH) analysis in mice by double-labelling the LHb with FISH probes for AADC in conjunction with FISH probes for the dopaminergic neuron marker tyrosine hydroxylase (TH), the serotonergic neuron marker tryptophan hydroxylase 2 (TPH2), or the glutamatergic neuron marker vesicular glutamate transporter 2 (VGLUT2). The majority of AADC mRNA colocalised with VGLUT2 mRNA (Fig. 1b, d) but not with TH or TPH2 mRNA (Fig. 1c, Supplementary Fig. 1), suggesting that a substantial majority of AADC-expressing neurons in the LHb are glutamatergic and non-monoaminergic.

To evaluate the pattern of acute stress-driven neuronal activity in the LHb-RMTg-VTA pathway, we measured c-Fos expression by FISH in the LHb, VTA and RMTg of mice following a 10-min tail suspension stress. c-Fos expression was increased in glutamatergic neurons, including D-neurons, in the mice exposed to acute stress compared with stress-naïve mice (Fig. 1e–i, Supplementary Fig. 2a). However, tail suspension-induced acute stress did not change the proportions of c-Fos-expressing glutamatergic neurons and D-neurons (Supplementary Fig. 2b, c). Moreover, the acute stress induced by the tail suspension stress increased c-Fos expression in VTA TH-positive neurons and decreased c-Fos expression in RMTg GAD1-positive neurons (Fig. 1j, k). Similar results were found after other acute stress conditions such as forced swimming and restraint (Supplementary Fig. 3). These findings indicate that acute stress attenuates RMTg GABAergic neuronal activity but augments VTA dopaminergic neuronal activity, despite the increase in LHb neuronal activity. Thus, LHb neurons may play an inhibitory role via an as-yet-unknown signalling pathway between the LHb and the RMTg (Fig. 1l).

### LHb D-neurons affect dopamine secretion via the RMTg-VTA
To investigate the function of LHb D-neurons in acute stress, we used chemogenetic manipulation in the AADC^Cre mouse model. First, we confirmed the coexpression of Cre recombinase with endogenous AADC expression in the LHb of AADC^Cre mice (Supplementary Fig. 4a–c). The selective chemogenetic activation of LHb D-neurons by CNO significantly decreased the immobility time in the TST (Fig. 2a, b) but did not affect locomotion in the open-field test (OFT; Supplementary Fig. 4d, e). FISH analysis revealed that the chemogenetic activation of LHb D-neurons increased c-Fos expression in VTA dopaminergic neurons while decreasing c-Fos expression in RMTg GABAergic neurons (Fig. 2c–h, Supplementary Fig. 5a, b). In addition, we showed that selective chemogenetic inhibition of LHb D-neurons during tail suspension stress resulted in enhanced c-Fos expression in the RMTg. This observation effectively negated the possibility of a depolarisation block phenomenon in the LHb D-neurons, which are known to display heightened activity under basal conditions (Supplementary Fig. 6). These findings demonstrate that LHb D-neurons can regulate the activity of RMTg GABAergic neurons and VTA dopaminergic neurons.

To determine whether VTA activation, driven by the chemogenetic stimulation of LHb D-neurons, promotes dopamine secretion, we directly measured the dopamine level in the ventral striatum, including the NAc (Fig. 2i, Supplementary Fig. 4f). Microdialysis analysis revealed that dopamine release was significantly increased in the ventral striatum immediately after the chemogenetic stimulation of LHb D-neurons (Fig. 2j).

### Reduced AADC in LHb neurons causes depressive-like behaviour
Next, we investigated whether AADC expression changes in rodents exposed to chronic stress, such as chronic restraint stress (CRS) or learned helplessness-inducing unpredictable electric foot shocks (US). Quantitative polymerase chain reaction (qPCR) analysis revealed that AADC mRNA was significantly decreased in rats exposed to CRS and in mice exposed to US compared with rodents that did not undergo stress (Fig. 3a, b). To examine the role of LHb AADC in the development of depressive-like phenotypes, we performed AADC knockdown in the LHb using adeno-associated virus (AAV)-mediated delivery of a short hairpin RNA (shRNA), sh-AADC (Fig. 3c). FISH analysis confirmed the loss of AADC mRNA in the LHb (Fig. 3d, e, Supplementary Fig. 7). Three weeks after the AAV-shRNA injection (Fig. 3f), AADC-knockdown mice showed delayed body weight gain (Fig. 3g) compared with vehicle-treated control mice and displayed depressive-like phenotypes, including despair-like behaviour and anhedonia in the TST and the sucrose preference test (SPT), respectively (Fig. 3h–j). However, AADC-knockdown mice showed normal anxiety-like behaviour and locomotion as tested by the OFT and elevated zero maze (EZM) test (Supplementary Fig. 8).

We next examined the pattern of c-Fos expression in both VTA dopaminergic neurons and RMTg GABAergic neurons in control and AADC-knockdown mice by FISH. In contrast to chemogenetic stimulation of LHb D-neurons (Fig. 2e–h), AADC knockdown reduced c-Fos expression in VTA dopaminergic neurons and increased c-Fos expression in RMTg GABAergic neurons (Fig. 3k–n, Supplementary Fig. 9a, b).

To examine whether the expression of AADC in the LHb is sufficient to prevent depressive-like behaviours, we selectively overexpressed mouse-derived AADC (mAADC) in the LHb by injecting AAV-double floxed-mAADC-EYFP or control AAV-double floxed-EYFP into the LHb of AADC^Cre mice (Supplementary Fig. 10c). Note that the Cre-dependent AAV expression was verified by detection of EYFP signals only in the HEK293T cells that were treated with AAV-double floxed-mAADC-EYFP (Supplementary Fig. 10a). Nineteen days after bilateral AAV injection, mice were subjected to US for 3 days. In mice that overexpressed AADC, US did not cause anhedonia-like or despair-like behaviour, and there was no difference in anxiety-like behaviours and

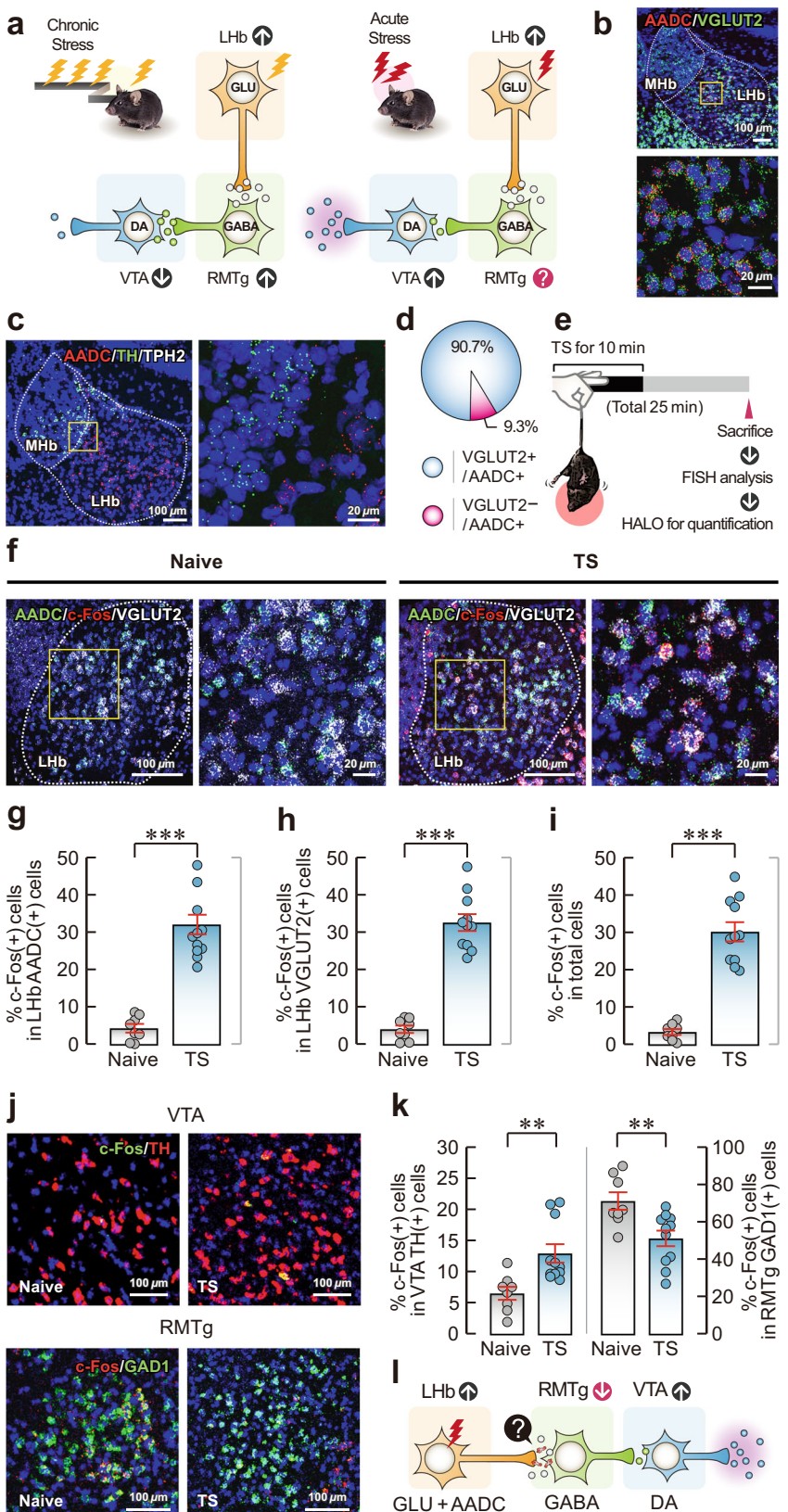

locomotion between mice that overexpressed AADC and control mice (Supplementary Fig. 10d–m). In contrast to the results from the AADC-knockdown mice, c-Fos expression was increased in VTA dopaminergic neurons and decreased in RMTg GABAergic neurons by AADC overexpression after chronic stress in mice that overexpressed AADC (Supplementary Fig. 11). These results together suggest that trace amines produced by AADC in LHb D-neurons may have a critical role in preventing depression.

### LHb D-neurons form synapses with RMTg GABAergic neurons

To examine the connectivity between LHb D-neurons and the RMTg, we used an anterograde tracing approach that identified which

**Fig. 1 | Acute stress alters the neuronal activity of the LHb-RMTg-VTA pathway.**
**a** Schematic of changes in neuronal activity of the LHb-RMTg-VTA pathway in
chronic and acute stress responses. DA, dopamine; GABA, gamma-aminobutyric
acid; GLU, glutamate. **b**, **c** Expression of AADC/VGLUT2 (**b**) and AADC/TH/TPH2 (**c**)
in the LHb by FISH. **d** Pie chart of the percentage of VGLUT2-expressing AADC-
positive cells in the LHb by FISH ($n = 3$ independent mice). **e**, Experimental sche-
matic of acute tail suspension (TS)-induced stress, followed by FISH analysis.
**f** Expression of AADC/c-Fos/VGLUT2 in the LHb of naïve and TS model mice by
FISH. **g–i** Percentages of c-Fos-expressing AADC-positive cells ($n = 8$ independent
Naïve mice and $n = 11$ independent TS mice) (**g**), c-Fos-expressing VGLUT2-positive
cells ($n = 8$ independent Naïve mice and $n = 11$ independent TS mice) (**h**) and c-Fos-
expressing cells in total cells in the LHb by FISH ($n = 8$ independent Naïve mice and

$n = 11$ independent TS mice) (**i**). **j** Expression of c-Fos/TH in the VTA (top) and c-Fos/
GAD1 in the RMTg (bottom) in naïve and TS model mice by FISH. **k** Percentages of c-
Fos-expressing TH-positive cells in the VTA (left; $n = 8$ independent Naïve mice and
$n = 11$ independent TS mice) and c-Fos-expressing GAD1-positive cells in the RMTg
(right; $n = 8$ independent Naïve mice and $n = 11$ independent TS mice) of naïve and
TS model mice. **l** Schematic of acute stress-mediated alterations in neuronal activity
in the LHb-RMTg-VTA pathway. Magnified images of the regions of interest (yellow
squares) are shown next to the main images in (**b**), (**c**) and (**f**). Unless otherwise
stated, statistical comparisons were performed using a two-tailed unpaired $t$ test.
**$p < 0.01$ and ***$p < 0.001$. Data are presented as the mean values ± s.e.m. Source
data are provided as a Source Data file.

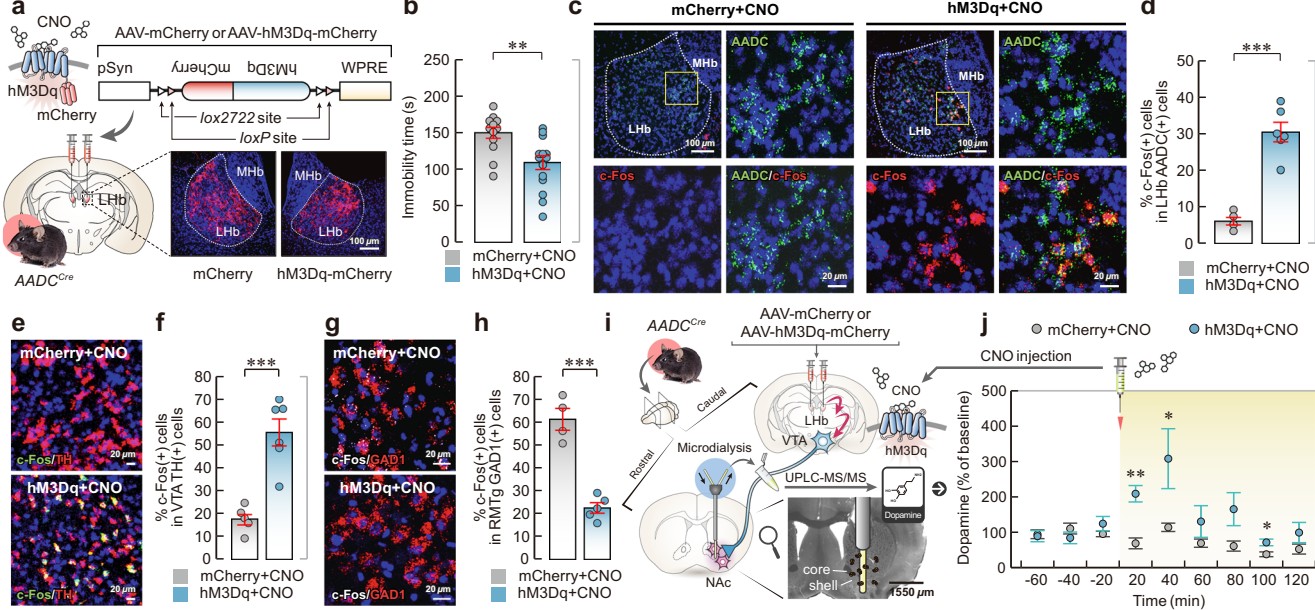

**Fig. 2 | Chemogenetic stimulation of LHb D-neurons regulates dopamine
secretion. a** Schematic of AAV vectors for Cre-dependent hM3Dq or mCherry
expression; representative images of the location of viral injection. **b** Effect of CNO-
induced activation of LHb D-neurons on behaviour observed in the TST as mea-
sured by immobility time ($n = 14$ independent mCherry mice and $n = 15$ indepen-
dent hM3Dq mice). **c–h** Changes in the expression of c-Fos in VTA dopaminergic
neurons and RMTg GABAergic neurons caused by chemogenetic stimulation of
LHb D-neurons. Representative FISH images in the LHb and medial habenula (MHb)
(**c**) and the percentage of c-Fos-expressing AADC-positive neurons in the LHb by
FISH ($n = 5$ independent mCherry mice and $n = 6$ independent hM3Dq mice) (**d**).
Expression of c-Fos/TH in the VTA (**e**) and c-Fos/GAD1 in the RMTg (**g**) by FISH.
Percentage of c-Fos-expressing TH-positive neurons in the VTA ($n = 5$ independent

mCherry mice and $n = 6$ independent hM3Dq mice) (**f**) and c-Fos-expressing GAD1-
positive neurons in the RMTg by FISH ($n = 4$ independent mCherry mice and $n = 5$
independent hM3Dq mice) (**h**). **i**, **j** Effect of chemogenetic stimulation of LHb
D-neurons on dopamine release in the NAc. Experimental schematic of micro-
dialysis (**i**) and quantification of dopamine-release changes pre- and postinjection
with CNO ($n = 7$ independent mCherry mice and $n = 5$ independent hM3Dq mice),
Two-tailed Mann–Whitney $U$ tests were used to compare differences between
groups (**j**). Magnified images of the regions of interest (yellow squares) are shown in
(**c**). Unless otherwise stated, statistical comparisons were performed using a two-
tailed unpaired $t$ test. *$p < 0.05$, **$p < 0.01$ and ***$p < 0.001$. Data are presented as
the mean values ± s.e.m. Source data are provided as a Source Data file.

neurons receive direct synaptic input from LHb D-neurons (Fig. 4a).
Axons from LHb D-neurons labelled with AAV-mGFP/synaptophysin-
mRuby mostly traversed through the fasciculus retroflexus and ter-
minated in the RMTg (Fig. 4b, c, Supplementary Movie 1).

To further confirm whether LHb D-neurons form direct synaptic
contacts, we assessed the activity of RMTg GABAergic neurons using
a combined optogenetic and electrophysiology approach (Fig. 4d).
Brief optogenetic stimulation of the terminals on LHb D-neurons
induced temporally precise inward currents that were abolished by
6-cyano-7-nitroquinoxaline-2,3-dione (CNQX) and amino-5-
phosphonopentanoic acid (AP5) application (Fig. 4e), suggesting
glutamatergic transmission. Thus, we next evaluated whether
D-neurons with downregulated AADC in the LHb can alter the
activity of RMTg GABAergic neurons. Current-clamp recordings on
RMTg GABAergic neurons of AADC-knockdown mice revealed a
significant increase in neuronal firing frequency and resting mem-
brane potential (RMP) compared with those of control mice

(Fig. 4f, g). These data indicate the inhibiting effect of LHb
D-neurons on RMTg GABAergic neuron activity.

## LHb D-neurons suppress RMTg GABAergic neuron activity
To further address whether selective stimulation of LHb D-neurons in
chemogenetically stimulated mice suppresses RMTg GABAergic
neuron activity, we measured the firing frequency of RMTg
GABAergic neurons using whole-cell patch-clamp recordings with
optogenetic stimulation. For optogenetic stimulation, an AAV
encoding either sh-AADC/ChR2 (channelrhodopsin-2) or GFP/ChR2
was bilaterally injected into the LHb of $AADC^{Cre}$ mice (Fig. 5a). In the
GFP/ChR2-expressing control mice, optogenetic stimulation of the
nerve terminals of LHb D-neurons in the RMTg significantly reduced
the spontaneous firing frequency of RMTg GABAergic neurons
(Fig. 5b, Supplementary Fig. 12a, b). However, this reduction in firing
frequency was not observed in AADC-knockdown mice (Fig. 5b,
Supplementary Fig. 12a, c). These data indicate that glutamatergic

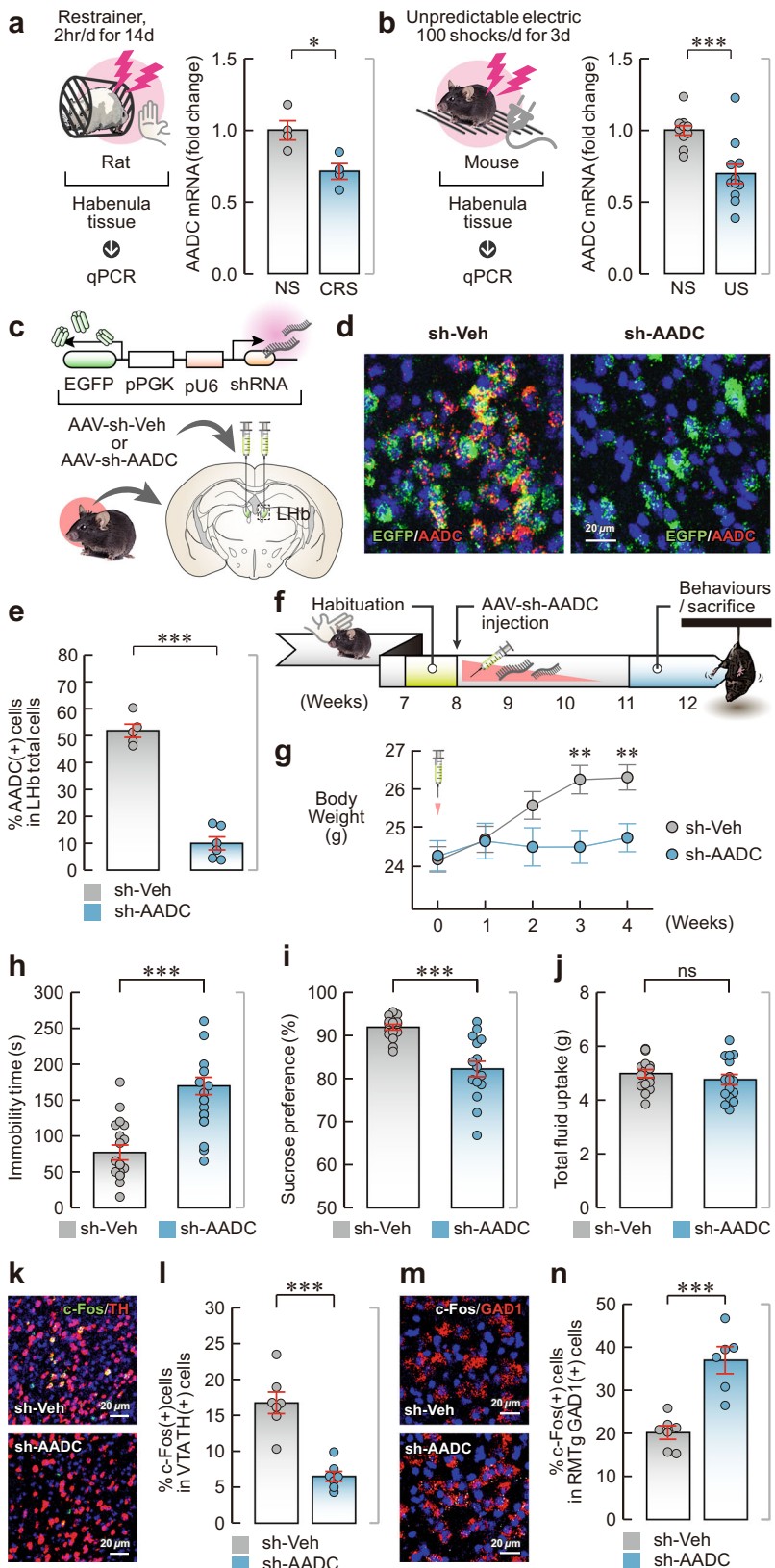

LHb D-neurons have an inhibitory effect on RMTg GABAergic neurons.

Given that the selective activation of D-neurons in the LHb plays an inhibitory role, we next explored the molecular pathway involved in the suppression of RMTg GABAergic neurons mediated by D-neurons. As D-neurons produce trace amines but no monoamines,

trace aminergic signalling may play a key role in the neural circuit from LHb D-neurons to RMTg GABAergic neurons. Because TAAR1 is the primary receptor for trace amines in the brain[32], we examined the expression of TAAR1 mRNA in the RMTg using FISH. Approximately 98% of GAD1-positive GABAergic neurons in the RMTg expressed TAAR1 mRNA (Fig. 5c, d, Supplementary Fig. 13),

**Fig. 3 | AADC knockdown in the LHb elicits depressive-like behaviours. a, b** Changes in expression of AADC mRNA in rodents exposed to CRS and US. Experimental schematics (**a** left, **b** left) and qPCR analysis for rats exposed to CRS (**a** right; $n = 4$ independent NS rats and $n = 4$ independent CRS rats, Two-tailed Mann–Whitney U-tests were used to compare differences between groups) and for mice exposed to US (**b** right; $n = 11$ independent NS mice and $n = 6$ independent US mice). NS, not stressed. **c**, Schematic, and location of the injection of AAV engineered to overexpress shRNA against AADC. Veh vehicle. **d, e** In vivo validation of AADC knockdown demonstrated by the expression of EGFP/AADC in the LHb by FISH (**d**), and the percentage of AADC-positive cells in the LHb by FISH ($n = 5$ independent sh-Veh mice and $n = 6$ independent sh-AADC mice) (**e**). **f** Experimental paradigm for behavioural assays. **g–j** Effect of AADC knockdown in the LHb on animals' body weight ($n = 11$ independent sh-Veh mice and $n = 11$ independent sh-

AADC mice) (**g**), immobility time in the TST ($n = 18$ independent sh-Veh mice and $n = 18$ independent sh-AADC mice) (**h**), percentage of sucrose preference (**i**) and total fluid uptake (**j**) in the SPT ($n = 16$ independent sh-Veh mice and $n = 16$ independent sh-AADC mice). **k–n** Effect of AADC knockdown in the LHb on VTA dopaminergic and RMTg GABAergic neuronal activity. Expression of c-Fos/TH in the VTA (**k**) and c-Fos/GAD1 in the RMTg (**m**) by FISH. Percentage of c-Fos-expressing TH-positive neurons in the VTA ($n = 7$ independent sh-Veh mice and $n = 7$ independent sh-AADC mice) (**l**) and c-Fos-expressing GAD1-positive neurons in the RMTg ($n = 7$ independent sh-Veh mice and $n = 6$ independent sh-AADC mice) (**n**) by FISH. Unless otherwise stated, statistical comparisons were performed using a two-tailed unpaired $t$ test. *$p < 0.05$, **$p < 0.01$ and ***$p < 0.001$. ns, not significant. Data are presented as the mean values ± s.e.m. Source data are provided as a Source Data file.

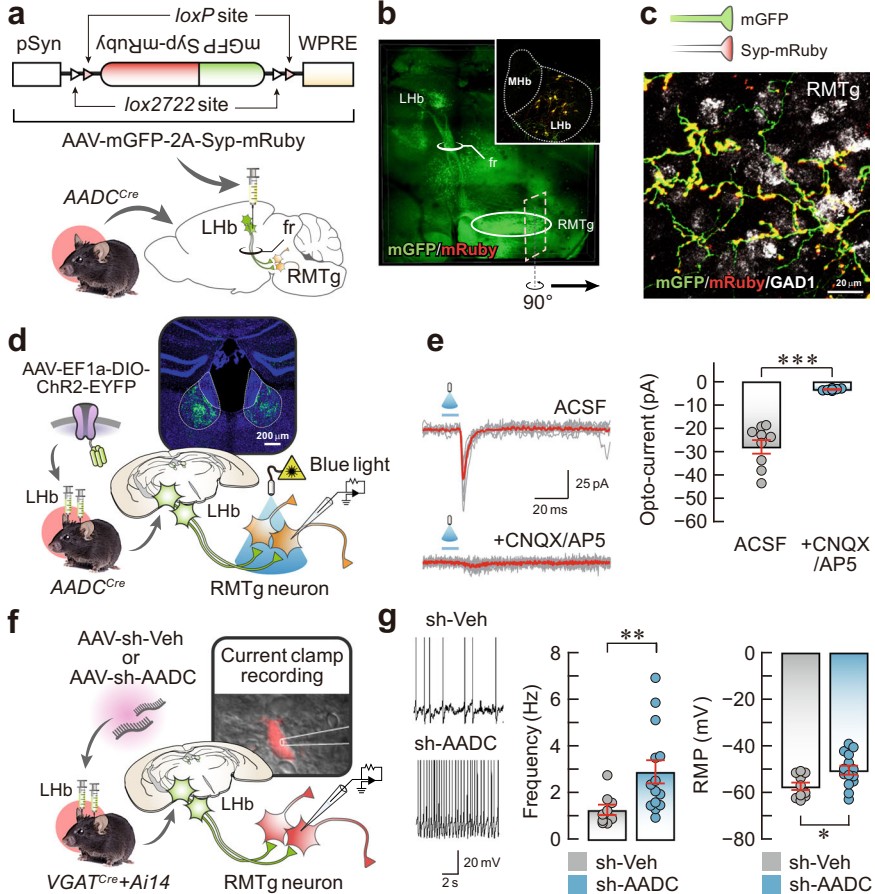

**Fig. 4 | LHb D-neurons innervate RMTg GABAergic neurons. a** Schematic of the AAV used and experimental procedure for anterograde tracing. **b** Representative AAV-hSyn-mGFP-Syp-mRuby infection pattern in the LHb (inset) and the sagittal fluorescent image showing LHb D-neuron projections. **c**, A rotated and magnified image of the region of interest (dashed square) from (**b**). LHb D-neuronal terminals on GABAergic neurons of the RMTg were detected by FISH. **d** Schematic of AAV-ChR2 injection into the LHb and the response of RMTg GABAergic neurons to brief optical stimulation. **e** Traces from oEPSCs (left) and summary data on normalised oEPSC amplitude under baseline conditions after CNQX (20 μM) and AP5 (50 μM)

perfusion in ACSF (right) ($n = 9$ cells from independent ChR2 mice). **f** Schematic of the method for measuring the firing rate of RMTg GABAergic neurons in AADC-knockdown $VGAT^{Cre}$::Ai14 mice and their vehicle controls. **g** Representative traces (left), firing frequency (middle; $n = 9$ cells from independent sh-Veh mice and $n = 14$ cells from independent sh-AADC mice) and RMP (right; $n = 10$ cells from independent sh-Veh mice and $n = 14$ cells from independent sh-AADC mice). Unless otherwise stated, statistical comparisons were performed using a two-tailed unpaired $t$ test. *$p < 0.05$, **$p < 0.01$ and ***$p < 0.001$. Data are presented as the mean values ± s.e.m. Source data are provided as a Source Data file.

suggesting that most RMTg GABAergic neurons could be affected by trace amines.

To assess the effect of trace amines on RMTg GABAergic neurons, brain slices containing the RMTg from control or AADC-knockdown mice were acutely exposed to the selective TAAR1 agonist RO5263397 ((S)−4-(3-fluoro-2-methylphenyl)−4,5-dihydrooxazol-2-amine) or to the antagonist EPPTB (N-(3-ethoxyphenyl)−4-pyrrolidin-1-yl-3-trifluoromethyl-benzamide) by bath application (Fig. 5e). RO5263397 (500 nM) application suppressed the

spontaneous firing frequency of RMTg GABAergic neurons in control mice. In contrast, the application of EPPTB (1 μM) rescued the RO5263397-mediated firing deficiency. Interestingly, the EPPTB treatment greatly increased the firing frequency over the basal level (Fig. 5f, Supplementary Fig. 12d, e). In AADC-knockdown mice, RO5263397 also significantly decreased the firing frequency of RMTg GABAergic neurons, which was restored to the basal level by EPPTB application (Fig. 5f, Supplementary Fig. 12d, f). Because the effect of RO5263397 on firing frequency in AADC-knockdown mice

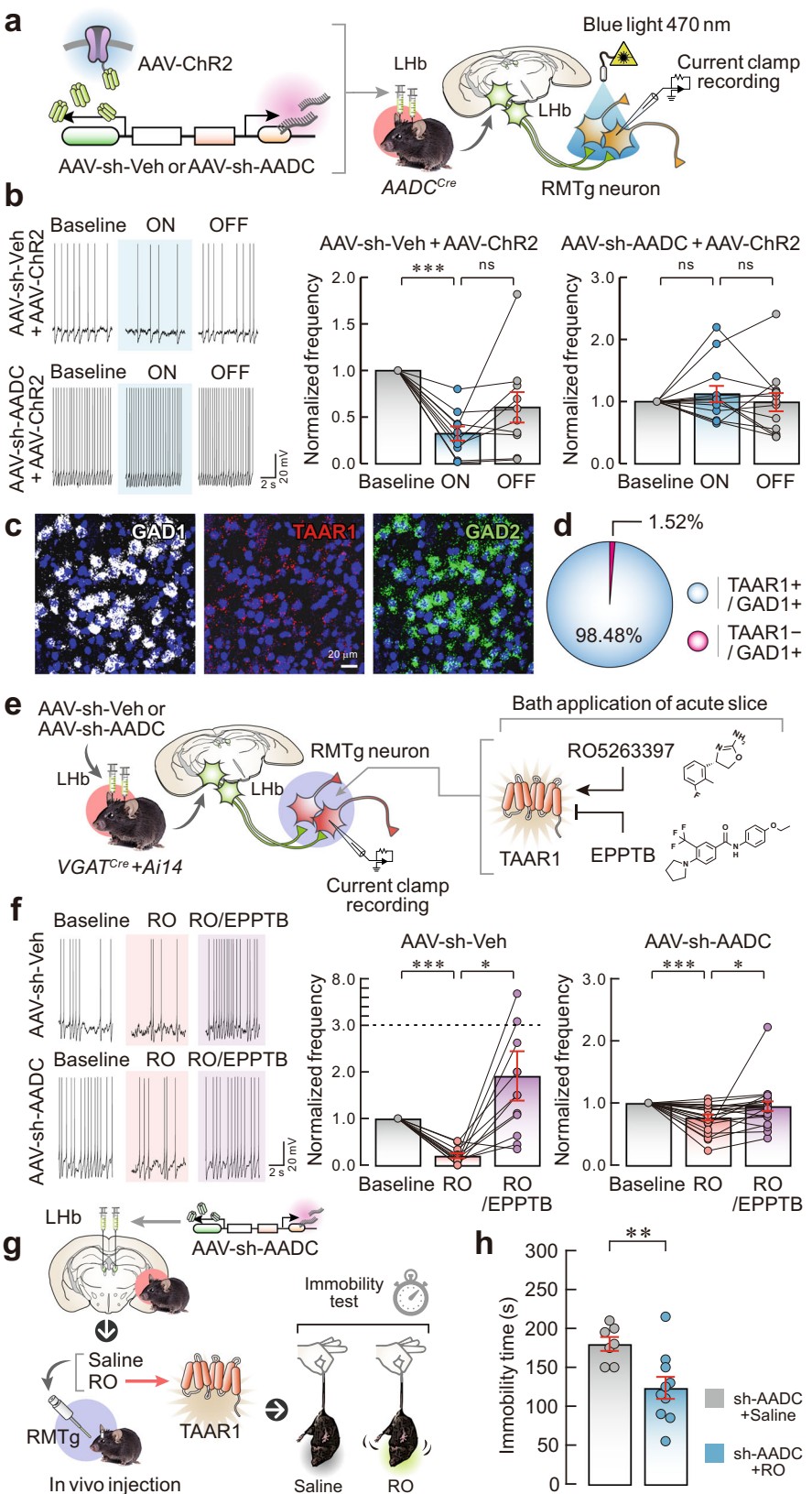

was weaker than that of control mice (Fig. 5f), we suspected that the level of TAAR1 expression in the RMTg GABAergic neurons might have changed in AADC-knockdown mice. However, FISH analysis revealed that there was no significant difference in TAAR1 expression between control and AADC-knockdown mice (Supplementary Fig. 14a, b). Moreover, we could not find any alterations in TAAR1

expression in the RMTg from the mice that underwent acute stress or US, regardless of the test used (Supplementary Fig. 14c–f). Given this constant and stable RMTg TAAR1 expression, the diminished RMTg responsiveness to RO5263397 in AADC-knockdown mice may not be caused by the altered level of TAAR1 in the RMTg but could be a consequence of an unknown effect originating from long-term

**Fig. 5 | LHb D-neurons suppress RMTg GABAergic neurons through trace aminergic signalling. a** Schematic of the AAV used and RMTg GABAergic neuronal activity induced by the optogenetic stimulation of LHb D-neurons. **b** Representative traces (left) and firing frequency in the control mice (middle; $n = 10$ cells from independent sh-Veh/ChR2 mice) and AADC-knockdown mice (right; $n = 13$ cells from independent sh-AADC/ChR2 mice) measured at baseline and in the ON or OFF phases of blue-light photoactivation. **c, d** Expression of GAD1/TAAR1/GAD2 mRNA in GABAergic neurons of the RMTg by FISH (**c**), and the percentage of TAAR1-expressing GAD1-positive GABAergic neurons in the RMTg by FISH ($n = 3$ independent mice) (**d**). **e** Schematic of AAV injection and RMTg GABAergic neuronal firing rates in response to the application of RO5263397 (RO) or EPPTB. **f** Representative traces (left) and firing frequency in control (middle;

$n = 11$ cells from independent sh-Veh mice) and AADC-knockdown mice (right; $n = 21$ cells from independent sh-AADC mice) measured in response to the application of RO5263397 (500 nM) or EPPTB (1 μM). The firing frequency was normalised to the baseline value (**b, f**). **g, h** Effect of RO5263397 application in AADC-knockdown mice in vivo on their performance in the TST. Schematic of an AAV in vivo injection (**g**), and TST immobility time results ($n = 7$ independent sh-AADC/Saline mice and $n = 10$ independent sh-AADC/RO mice), a two-tailed unpaired $t$ test was used to compare differences between groups (**h**). Unless otherwise stated, statistical comparisons were performed using repeated measure one-way ANOVA followed by contrast test. $*p < 0.05$, $**p < 0.01$ and $***p < 0.001$. Data are presented as the mean values ± s.e.m. Source data are provided as a Source Data file.

AADC knockdown in the LHb that could change the RMTg cellular properties, as evidenced by our electrophysiology results (Fig. 4g).

To collectively elucidate the trace aminergic signalling of LHb D-neurons to the RMTg, we added EPPTB during terminal optogenetic stimulation of those D-neurons in brain slices that included the RMTg (Supplementary Fig. 15a). The recorded firing frequency in the RMTg decreased during optogenetic stimulation but increased after EPPTB treatment (Supplementary Fig. 15b–e). Furthermore, we found that pretreatment with EPPTB effectively blocked the impact of terminal optogenetic stimulation of LHb D-neurons (Supplementary Fig. 15f–i). To determine the effect of trace amines on RMTg GABAergic neurons in vivo, we administered RO5263397 or saline directly into the RMTg in AADC-knockdown mice and subjected them to the TST (Fig. 5g). The immobility time in the TST was reduced in RO5263397-treated AADC-knockdown mice compared with saline-treated AADC-knockdown mice (Fig. 5h). To further address whether trace aminergic signalling modulates the effects of acute stress on behaviour, we observed whether blockade of trace aminergic signalling through direct administration of EPPTB into the RMTg of naïve mice induced changes in immobility. The addition of EPPTB into the RMTg caused increased immobility compared with vehicle-treated mice, similar to the effect of AADC knockdown (Supplementary Fig. 16). These results suggest that trace amines produced by LHb D-neurons can sufficiently modulate RMTg GABAergic neurons in response to acute stress.

**Biphasic control of RMTg GABAergic neurons with acute stress**

Li et al. reported that RMTg firing frequency has an initial excitation phase upon initiation of restraint stress and then has a rebound inhibition phase that brings the firing frequency below baseline at the offset of restraint stress[33]. To determine whether trace aminergic signalling by LHb D-neurons can influence RMTg GABAergic responses to stressful events, we performed in vivo recordings before (-15–0 min), during (0–10 min) and after (10–30 min) tail suspension stress. To genetically identify GABAergic neurons in the RMTg using an optical-tagging strategy[34], control and AADC-knockdown *VGAT*$^{Cre}$ mice were injected with recombinant AAVs expressing ChR2 and implanted with optrodes in the RMTg (Fig. 6a). Among 137 cells in control mice and 82 cells in AADC-knockdown mice, 34 and 29 neurons, respectively, were classified as GABAergic neurons based on their optical sensitivity, which was measured by providing 10 trains of 10 blue-light pulses (each train 5 ms duration per pulse) at 30 Hz (Fig. 6b, Supplementary Fig. 17a, d). ChR2-expressing GABAergic neurons displayed a high probability of light-evoked spikes (≥0.81) and a short spike latency from the light onset (<5.1 ms) (Supplementary Fig. 17b, e). When the basal firing frequency of GABAergic neurons was compared, AADC knockdown in the LHb significantly increased GABAergic firing frequency (Supplementary Fig. 17g), which was consistent with the in vitro electrophysiological results (Fig. 4g). The majority of GABAergic neurons in control mice were significantly excited during the first 5-min period of tail suspension (94%, 32/34 cells) and inhibited during the 5-min period right after the stress (91%, 31/34 cells) (Fig. 6c–e and Supplementary Figs. 17c, 18a, b). However, AADC knockdown

substantially altered the proportions of biphasic responses (chi-square tests, $\chi^2(2) > 8.3$, $p < 0.05$), so that only 65.5% (19/29 cells) and 55% (16/29) of GABAergic neurons were excited during the stress and inhibited after the stress, respectively (Fig. 6f–h and Supplementary Figs. 17f, 18c, d). Interestingly, rebound inhibition was abolished in AADC-knockdown mice (Fig. 6h). Thus, these results indicate that LHb neurons initially excite GABAergic neurons in the RMTg by glutamatergic transmission at the onset of stress, but eventually inhibit them by the D-neuron-mediated trace aminergic signalling that immediately follows.

## Discussion

Suppression of hyperexcitability of LHb neurons has emerged as a major target for restoring the LHb-midbrain monoamine pathway to its normal condition. As part of this effort, several researchers have attempted to elucidate the existence and role of GABAergic neurons within the LHb[35–39]. However, the presence and function of GABAergic neurons capable of inhibiting LHb hyperexcitability were insufficient to form reliable conclusions. More importantly, because LHb D-neurons are also glutamatergic, we reveal that these cells can sufficiently perform initial excitatory functions using glutamates and rebound inhibitory functions using trace amines in the well-known LHb-RMTg-VTA pathway and that the latter inhibitory role plays a protective function under acute stress. This unexpected bidirectional role of the neural pathways (Fig. 7) may explain the many contradictory results obtained by other researchers.

Aversive external stimuli enhance neuronal activities of the LHb and the RMTg[17,33]. For instance, optogenetic or electrical stimulation of LHb neurons evokes excitatory postsynaptic currents (EPSCs) in RMTg GABAergic neurons[14,40,41]. However, most studies observed responsiveness only to the stimuli that occurred within a very short time period (a few seconds) and did not determine whether RMTg firing frequency changed with time after the stimulus onset (tens of minutes). Of note, Li et al.[33] performed in vivo recordings and showed that in response to stimuli such as restraint stress, the RMTg firing frequency was inhibited to below the baseline level for a considerable amount of time for approximately 3 min before the stress was offset. This reduction in RMTg firing frequency was also induced by tail suspension stress, whereas an increased RMTg firing frequency in the LHb in AADC-knockdown mice was maintained for approximately 7 min without rebound inhibition (Fig. 6). The LHb can induce a transient increase in RMTg firing frequency in the early stage of an aversive stimulus. But eventually, trace amine secretion by D-neurons leads to a reduction in RMTg firing frequency, which means that glutamate and trace amines from LHb D-neurons biphasically regulate the activity of RMTg GABAergic and VTA dopaminergic neurons. Contrary to our findings, previous studies have demonstrated that acute and repeated foot shocks (four, 0.5 mA, 0.5 s; or five, 0.5 mA, 0.8 s) induce upregulation of c-Fos expression in the core region of the RMTg[42,43]. Notably, a separate investigation indicated that a single foot shock (0.5 mA, 0.8 s or 4 s; or 0.8 mA, 0.8 s) did not lead to increased c-Fos expression in the RMTg[44]. Foot shocks are known to act as a potent stressor that can

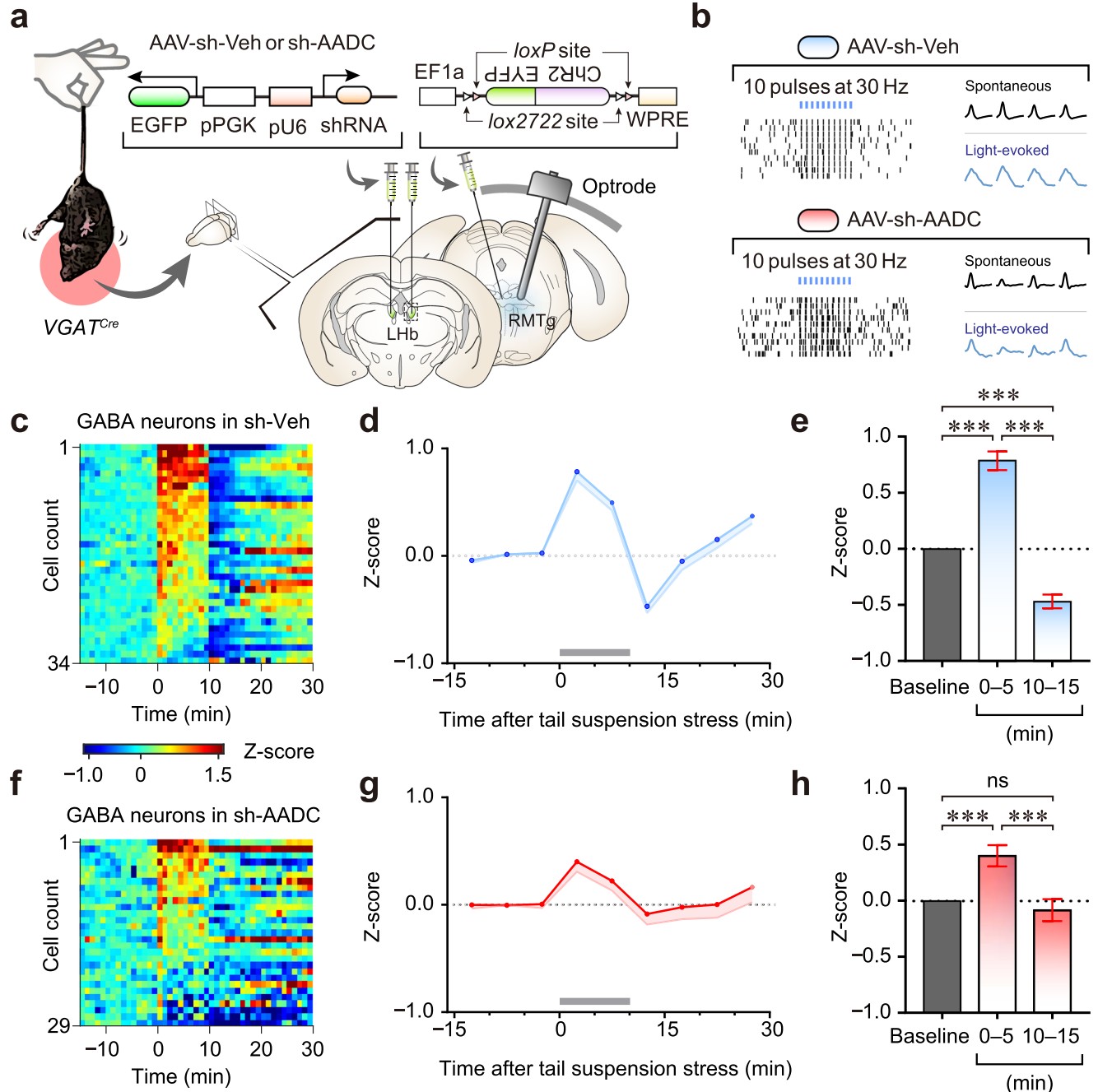

**Fig. 6 | LHb D-neurons regulate RMTg GABAergic responses to acute stress through trace aminergic signalling. a** Schematic of in vivo recording of RMTg GABAergic neuronal activity under tail suspension stress. **b** Optical identification of ChR2-expressing GABAergic neurons in the RMTg. The raster plots of two representative GABAergic neurons from control and AADC-knockdown mice. **c**, **f** Heatmaps of the average normalised firing of GABA neurons (5-min bins) in control (**c**) and AADC-knockdown (**f**) mice. Tail suspension stress was given for 10 min from time 0. **d**, **g** Normalised firing rates of all GABAergic neurons in control (**d**) and AADC-knockdown (**g**) mice. Tail suspension stress is shown as grey bars in (**d**) and (**g**). **e**, **h** Comparisons of average firing rates before (baseline), during (0–5 min) and after (10–15 min) tail suspension in control ($n = 34$ cells from independent sh-Veh mice) (**e**) and AADC-knockdown ($n = 29$ cells from independent sh-AADC mice) (**h**) mice. Unless otherwise stated, statistical comparisons were performed using repeated measure one-way ANOVA followed by contrast test. *$p < 0.05$, **$p < 0.01$ and ***$p < 0.001$. Data are presented as the mean values ± s.e.m. Source data are provided as a Source Data file.

elicit bursting firing in LHb neurons[17]. Such repetitive firing patterns, which have been observed in depression models[7], may lead to heightened glutamatergic input from the LHb to the RMTg in comparison to trace aminergic input. This suggests that the expression of c-Fos in the RMTg may vary depending on the type and intensity of stress, with repetitive high-intensity stress likely to cause alterations in AADC expression within the LHb.

When LHb neurons are excited by an external stimulus, the relatively slow secretion and intracellular penetration rates of trace amines

appear to subsequently cause the inhibition of the increased firing frequency of RMTg GABAergic neurons that is normally caused by the rapid action of increased glutamate. Unlike their structural neurotransmitter analogues, trace amines do not appear to be stored. Most trace amines diffuse across plasma lipid bilayers in the absence of a transporter, with diffusion half-lives of 15 s or less[45]. Furthermore, the release of trace amines is not increased by $K^+$-induced depolarisation observed in brain slices[46], suggesting that these compounds are neither stored in synaptic vesicles nor released by exocytosis, presumably

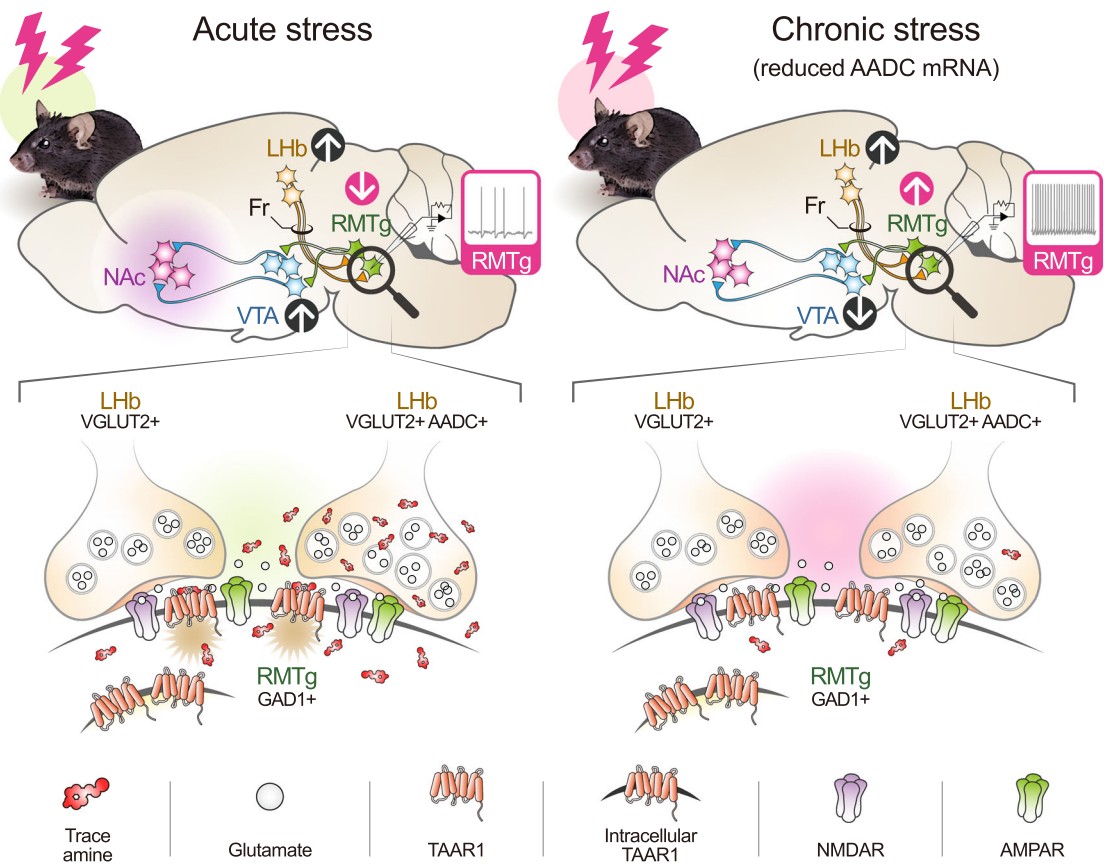

**Fig. 7 | Schematic illustration of the suggested role of LHb D-neurons.** LHb D-neurons project to RMTg GABAergic neurons and are activated by acute stress. Acute stress evokes trace amine release from LHb D-neurons to RMTg GABAergic neurons, leading to neuronal inactivation via TAAR1-mediated trace aminergic signalling. As the effect of trace amines overrides the excitatory glutamatergic transmission, LHb D-neurons act as the negative regulator of RMTg GABAergic neurons. This action ultimately activates VTA dopaminergic neurons to promote dopamine secretion into the NAc. In a depressed brain induced by chronic stress, the weakened trace aminergic signalling suppresses the activity of VTA dopaminergic neurons.

indicating 'release' by simple diffusion across the membrane. Contrary to prior expectations, our study revealed that stimulation of LHb D-neuron terminals via optogenetics resulted in trace amine release, leading to a reduction in firing frequency within the RMTg (Fig. 5 and Supplementary Fig. 15). These results suggest that depolarisation of D-neurons in the LHb may trigger trace amine release, which, in turn, can activate TAAR1 in the RMTg. This plausible molecular mechanism may account for the observed late-phase inhibition of D-neurons. TAAR1 is a G protein-coupled receptor expressed in the central nervous system. TAAR1 is coupled to Gαs-signalling pathways, which promotes cyclic adenosine monophosphate production via stimulation of adenylyl cyclase[47]. In addition, TAAR1-mediated reduction in the firing rate of VTA dopaminergic neurons appears to occur subsequent to activation of G protein-coupled inwardly rectifying potassium channels[24]. The TAAR1 agonist RO5263397 suppresses the firing rate of VTA dopaminergic neurons and dorsal raphe nucleus serotonergic neurons; this suppression is reversed by the TAAR1 antagonist EPPTB, consistent with the activation of RMTg GABAergic neurons (Fig. 5f). Hence, because the speed of G protein-coupled receptor-mediated signals is slower than that of NMDA- and AMPA-type glutamate receptors[48], we propose that LHb D-neurons, which are polyphenotypic, may have distinct functions with regard to biphasic regulation in the RMTg.

Previous studies have highlighted possible bidirectional roles of VTA dopamine neurons under stress conditions. In unpredictable chronic mild stress, the activity of VTA dopaminergic neurons is reduced, leading to reduced dopamine secretion. Chronically stressed mice show depressive-like behaviours, and these impaired behaviours are restored by phasic optogenetic activation of VTA dopaminergic neurons[49]. In contrast to results from animal models of depression after unpredictable chronic mild stress, mice susceptible to chronic social defeat stress display higher neuronal activity in the VTA compared with control and unsusceptible mice[50]. Chaudhury et al. have also shown that phasic optogenetic activation of VTA dopaminergic neurons induces a susceptible phenotype as measured by the social interaction test and the SPT[51]. To explain these conflicting results, we compared AADC expression in the LHb in chronic social defeat stress-susceptible and control mice and found no differences. Because chronic social defeat stress activates LHb and lateral hypothalamus (LH) neurons (afferent neurons of the LHb)[52] and because LHb D-neurons lowered the activity of RMTg GABAergic neurons (our results), we expect that continuous activation of VTA dopaminergic neurons can be maintained by RMTg inhibition through trace aminergic signalling of D-neurons. Indeed, activation of the LH-LHb circuit alleviates social avoidance and despair-like and anxiety-like behaviours after chronic social defeat stress[52].

There remain several outstanding issues that require further investigation. Notably, we observed two key differences between control and AADC-knockdown mice. Firstly, the attenuation of endogenous trace amine secretion from nerve terminals of the LHb resulting from AADC knockdown led to a reduction in responsiveness to RO5263397 in the RMTg (Fig. 5f). Secondly, LHb AADC knockdown resulted in a lower magnitude of excitation during the first 5-min period of tail suspension compared with control mice (Supplementary Fig. 18a and c). These observations suggest that the long-term effects of AADC knockdown may alter cellular responsiveness. Electrophysiological data indicated

that AADC knockdown in the LHb increased the basal firing frequency and RMP in RMTg GABAergic neurons (Fig. 4g, Supplementary Fig. 17g), which suggests altered signal transduction through TAAR1 G proteins. Notably, the majority of LHb neurons (67.6%) affect the RMTg[14], and it is known that LHb neurons project to the posterior VTA. It would be of interest to confirm the proportion of these neurons that are D-neurons. In this study, female mice were excluded from the experiments due to the potential impact of the oestrous cycle on the stress response, anxiety, and depression, as previously reported[53,54]. Therefore, future studies are warranted to gain a better understanding of the functional features of the cellular and molecular changes to TAAR1-mediated signalling for the monoaminergic RMTg pathway, including investigations in female mice.

Overall, the results from our study offer new insights into the role of the LHb and demonstrate that glutamatergic neurons are involved in a biphasic regulation pattern that is both excitatory and inhibitory. This mechanism involves a transient increase in glutamate transmission induced by external stimuli and subsequent inhibition through a change in membrane properties by a continuous increase in trace aminergic signalling. These findings in trace aminergic signalling in the LHb could lead to a broader understanding of potential therapies for depression.

## Methods

### Animals
Adult male C57BL/6J mice and adult male Sprague–Dawley rats (7 weeks of age) were purchased from Japan SLC, Inc. (Hamamatsu, Japan). *AADC^Cre* mice (B6.FVB(Cg)-Tg(Ddc-cre)SD56Gsat/Mmucd, RRID:MMRRC_037410-UCD) were obtained from the Mutant Mouse Resource and Research Center of the University of California, Davis (Davis, CA, USA) and were used for chemogenetic, optogenetic, microdialysis and anterograde tracing experiments. *VGAT^Cre* mice (B6J.129S6(FVB)-Slc32a1^tm2(cre)Lowl/MwarJ, RRID:IMSR_JAX:028862)[55] were crossed with Ai14 mice (B6.Cg-Gt(ROSA)26Sor^tm14(CAG-tdTomato)Hze/J, RRID:IMSR_JAX:007914)[56] to identify GABAergic neurons in the RMTg. All animals were housed three to four per cage under a 12/12-h light/dark cycle (lights on at 8 a.m.) and given ad libitum access to food and water. Animals were habituated to the facility for at least 1 week prior to the experiments. All animal experiments were approved by the Korea University Institutional Animal Care and Use Committee (IACUC) and performed in accordance with the guidelines of Korea University (study approval number KOREA-2017-0007-C1).

### FISH analysis
In situ hybridisation was performed in accordance with the procedure described by Yang et al.[57] to measure changes in mRNA expression. In brief, brains were quickly removed from euthanized experimental mice, and frozen brain sections (14-μm thick) were obtained by coronal cutting through the Hb, VTA, and RMTg. Sections were thaw-mounted onto Superfrost Plus Microscope Slides (12-550-15, Thermo Fisher Scientific, Waltham, MA, USA). Then, the sections were fixed in 4% paraformaldehyde for 10 min, dehydrated in increasing concentrations of ethanol for 5 min and air-dried. Tissues were then pretreated by protease digestion for 10 min at room temperature. Probe hybridisation and amplification were performed at 40 °C using a HybEZ hybridisation oven (Advanced Cell Diagnostics, Newark, CA, USA). The probes used in this study are described in Supplementary Table 1. The labelled probes were conjugated to Alexa Fluor 488, Atto 550 or Atto 647. The sections were hybridised with the labelled probe mixture at 40 °C for 2 h. Unbound hybridisation probes were removed by washing the sections three times with 1× wash buffer at room temperature for 2 min; then, the slides were treated with Amplifier 1-FL for 30 min, Amplifier 2-FL for 15 min, Amplifier 3-FL for 30 min and Amplifier 4 Alt B-FL for 15 min. After each amplifier treatment, the sections were washed with 1× wash buffer at room temperature for 2 min before

being treated with the next amplifier solution. Next, the slides were viewed, analysed and photographed using a TCS SP8 dichroic/CS microscope (Leica, Wetzlar, Germany). After FISH was performed, the mean number of dots per cell and per unit area (μm²) were quantified using the HALO image analysis algorithm in HALO v2.3.2089.18 software (Indica Labs, Albuquerque, NM, USA)[58].

### Histology
Mice were anaesthetized with a mixture alfaxalone (40 mg per kg): xylazine (10 mg per kg) and perfused with 0.9% saline and then with 4% paraformaldehyde. The brains were removed from the skulls and postfixed in 4% paraformaldehyde overnight at 4 °C. After postfixation, the brains were incubated in 30% sucrose at 4 °C. The brains were cut into 100-μm-thick coronal sections with a cryotome (CM300, Leica); sectioned brain regions encompassed the habenula. Next, sections were washed with phosphate-buffered saline (PBS) three times for 5 min at room temperature. The washed sections were incubated with Hoechst 33342 (H3570, Thermo Fisher Scientific), a blue fluorescent stain that binds double-stranded DNA, at room temperature for 10 min. The stained sections were immersed in mounting solution for 30 min at 37 °C. The sections were viewed and photographed using a TCS SP8 dichroic/CS microscope (Leica).

### Immunohistochemistry
The brains were prepared using the same method as in histology and cut into 40-μm-thick coronal sections with a cryotome (CM300, Leica). The sectioned brain regions encompassed the habenula and the RMTg. Brain sections were incubated in primary antibodies for 1 h at 37 °C. The primary antibodies were diluted in PBS containing 3% bovine serum albumin (BSA) and 0.2% Triton X-100. The following primary antibodies were used: anti-c-Fos antibody (1:500; #SC-52G, Santa Cruz Biotechnology, Inc., Dallas, TX, USA) and anti-GABA antibody (1:500; #A2052, Sigma-Aldrich, St. Louis, MO, USA). Next, the sections were washed with PBS and incubated in a cocktail of Alexa Fluor® 488 donkey anti-goat (1:500; #A-11055, Thermo Fisher Scientific) or Alexa Fluor® 647 donkey anti-rabbit (1:500; #711-605-152, Jackson Immuno Research Inc., West Grove, PA, USA) conjugated secondary antibodies in PBS containing 3% BSA and 0.2% Triton X-100 for 2 h at room temperature. The secondary antibody was washed with PBS and further incubated with Hoechst (1:1000; #H3570, Thermo Fisher Scientific) at room temperature for 10 min. The sections were immersed in mounting solution and images were captured using a TCS SP8 dichroic/CS microscope (Leica).

### Stress models
Acute stress was induced in mice by tail suspension, forced swimming and restraint stress[18,33,44]. After 1 week of habituation to their home cages, experimental mice were suspended by the tail for 10 min, placed in a clear cylinder filled with 24–25 °C water for 6 min and then placed in a well-ventilated 50 mL polypropylene conical tube (54050, SPL Life Sciences Co., Pocheon, South Korea) for 6 min. The mice were sacrificed 25 min after the onset of acute stress, and the brains were collected for FISH analysis.

For CRS model[59], 8-week-old Sprague–Dawley rats were subjected to restraint stress in DecapiCones® rodent restrainers (DC-200, Braintree Scientific, Inc., Braintree, MA, USA) for 2 h per day for 2 weeks. Control rats that underwent no stress were maintained in their home cages without being disturbed.

For US model[60], 8-week-old C57BL/6J mice were placed in shock chambers of dimensions 30-cm wide × 30-cm deep × 25-cm high (Multi Conditioning System, TSE Systems, Inc., Chesterfield, MO, USA), where they underwent 100 inescapable electric foot shocks of 0.3-mA intensity and 5-s duration with intershock intervals of 5–99 s.

## qPCR

qPCR was performed using the same protocol as a previous study[59]. Briefly, brains were quickly removed from euthanized from rats (10 weeks of age) and mice (8 weeks of age), and the habenula was carefully dissected. The RNA from the habenula tissue was isolated and subjected to reverse transcription. The reverse-transcribed sequences were then amplified by qPCR, and the products were electrophoresed using 2% agarose gels. The comparative ΔΔCt method was used for the relative quantification of the amplification products and to calculate the fold changes in gene expression between rats that underwent CRS and those that had no stress[61] and between naïve mice and those that that exposed to learned helplessness-inducing unpredictable electric foot shocks. The expression levels of AADC were normalised to the expression level of GAPDH, encoded by the housekeeping gene *Gapdh*. The sequences for the specific primers were as follows. For the CRS rat model, AADC forward, 5′-TTCTTCGCTTACTTCCCCACG-3′; AADC reverse, 5′-CCCAGCCAATCCATCATCACT-3′; GAPDH forward, 5′-CAT CCACTGGTGCTGCCAAGGCTG-3′; and GAPDH reverse, 5′-ACAACCTG GTCCTCAGTGTAFCCCA-3′. For the LH mouse model, AADC forward, 5′-GGCTTACATCCGAAAGCACG-3′; AADC reverse, 5′-CTTTAGCCGGAA GCAGACCA-3′; GAPDH forward, 5′-ACCCAGAAGACTGTGGATGG-3′; and GAPDH reverse, 5′-CACATTGGGGGTAGGAACAC-3′.

## Viruses

The virus production methodology used in this study to achieve AADC knockdown was based on a previously described protocol[59] but was modified in that we used recombinant AAV serotype 2/9 and the virus titres were determined by real-time PCR. We used AAV plasmids expressing hM3Dq (pAAV-hSyn-DIO-hM3D(Gq)-mCherry; 44361, Addgene, Watertown, MA, USA), hM4Di (pAAV-hSyn-DIO-hM4Di-mCherry; 44362, Addgene), mCherry (pAAV-hSyn-DIO-mCherry; 50459, Addgene), mGFP/synaptophysin-mRuby (pAAV-hSyn-FLEx-mGFP-2A-Synaptophysin-mRuby; 71760, Addgene) and ChR2 (pAAV-EF1a-double floxed-hChR2(H134R)-EYFP-WPRE-HGHpA; 20298, Addgene). The viral vector was constructed using GFP (pAAV-U6-GFP; purchased from Cell Biolabs, Inc., San Diego, CA, USA) with the shRNA sequence against AADC mRNA (sh-AADC, 5′-GTGATCTAGCAAG-CAGTGT-3′) inserted to knock down AADC expression. A virus encoding only GFP and with no shRNA was used as a control. For AADC-knockdown validation, the full-length complementary DNA (cDNA) of *AADC* (GenBank accession no. NM_001190448) was obtained from a C57BL/6N mouse brain using reverse transcription-PCR based gateway cloning method (Thermo Fisher Scientific, Inc., Waltham, MA, USA) and cloned into a pDEST-CMV-C-EGFP destination vector, (122844, addgene) by gateway cloning. For transfection, HEK293T cells were plated on 60-mm dishes and cultured for 24 h. Plasmids encoding GFP-AADC, AADC or scrambled shRNAs were transfected into the cells using Lipofectamine (Thermo Fisher Scientific, Inc., Waltham, MA, USA). Seventy-two hours after transfection, HEK293T cells were suspended in radioimmunoprecipitation assay lysis buffer (BioPrince, Chuncheon, South Korea) containing a protease inhibitor cocktail (Roche, Basel, Switzerland). After centrifugation, cell lysates were cleared of cell debris, and the remaining components were separated by 10% sodium dodecyl sulfate–polyacrylamide gel electrophoresis before being transferred to polyvinylidene fluoride membranes. Membranes were immunoblotted overnight at 4 °C with a primary anti-GFP (B-2) antibody (1:1000; sc-9996, Santa Cruz Biotechnology, Inc.). The membranes were washed, incubated with species-specific horseradish peroxidase-conjugated secondary antibody for 1 h at room temperature and developed using electrochemiluminescence solution (Thermo Fisher Scientific). To overexpress AADC in LHb D-neurons, we made a Cre-dependent AADC-expressing AAV-AADC viral vector. For this, the pAAV-EF1a-double floxed-mAADC-EYFP overexpression plasmid was constructed by replacing the hChR2 region of the pAAV-EF1a-double floxed-hChR2(H134R)-EYFP-WPRE-HGHpA vector with mouse AADC (*mAADC*) sequence. For mAADC-overexpressing AAV, *mAADC* was cloned into pAAV-EF1a-double floxed-hChR2(H134R)-EYFP-WPRE-HGHpA using the GeneArt™ Gibson Assembly® HiFi Cloning Kit (A46624, Thermo Fisher Scientific). Overexpression of pAAV-EF1a-double floxed-mAADC-EYFP in HEK293 cells showed Cre-dependent fluorescence. Cells expressing mAADC-EYFP fluoresced green (Supplementary Fig. 10a). The expression of mAADC-EYFP fusion protein was confirmed with anti-DOPA decarboxylase (1:1000; ab3905, Abcam, Cambridge, United Kingdom), anti-GFP (1:5000; ab290, Abcam) and β-actin (1:5000; sc-47778, Santa Cruz Biotechnology, Inc.) antibodies by western blot.

## Stereotactic surgery

Adult male mice (8 weeks of age) were anaesthetised with isoflurane (5% induction, 1% maintenance) and placed on a stereotactic apparatus (Ultra Precise Mouse Stereotaxic Instruments; Stoelting Co., Wood Dale, IL, USA). After an incision was made in the scalp, a craniotomy was performed using a hand drill so that the virus designed to achieve AADC knockdown could be injected into the LHb. Approximately 1 μL virus was injected into the LHb (coordinates from bregma: −1.58 mm anterior/posterior (A/P), ±0.9 mm medial/lateral (M/L), −3.1 mm dorsal/ventral (D/V) and 10° angle towards the midline in the coronal plane) using a 30-gauge microinjection cannula (P1 Technologies, Roanoke, VA, USA) and an UltraMicroPump III (World Precision Instruments, Sarasota, FL, USA) at 0.12 μL/min. The incision was closed with 9-mm autoclips (205016, MikRon Precision, Inc., Gardena, CA, USA), and antibiotics and analgesics were administered to the mice. Mice were placed in a clean cage on a heating pad and allowed to recover from anaesthesia. Then, they were kept in their home cage for 3 weeks so that the AAV could take effect before behavioural tests were performed.

For chemogenetic stimulation experiments, recombinant AAVs expressing hM3Dq or mCherry were bilaterally injected into the LHb of *AADC^Cre* mice. A CMA 7 guide cannula (CMA Microdialysis, Kista, Sweden) was implanted to measure dopamine in the ventral striatum (+1.00 mm A/P, +1.60 mm M/L and −3.00 mm D/V from bregma), including the NAc. Mice expressing hM3Dq were injected with clozapine N-oxide (CNO; BML-NS105, Enzo Life Sciences, Inc., Farmingdale, NY, USA), prepared in sterile 1× PBS with 0.5% dimethyl sulfoxide (DMSO; D8418, Sigma-Aldrich), at 5 mg kg body weight 40 min before the start of behavioural testing.

For chemogenetic inhibition experiments, recombinant AAVs expressing hM4Di or mCherry were bilaterally injected into the LHb of *AADC^Cre* mice. Mice expressing hM4Di were injected with CNO at 5 mg kg body weight 40 min before the start of tail suspension stress, and they were perfused with 0.9% saline and then with 4% paraformaldehyde for IHC 25 min after the onset of the stress.

For knockdown experiments, recombinant AAVs expressing GFP or sh-AADC were bilaterally injected into the LHb of wild-type mice. A 26-gauge guide cannula (P1 Technologies) was implanted to enable the administration of the TAAR1 agonist RO5263397 in the RMTg (−4.24 mm A/P, +0.70 mm M/L, −4.00 mm D/V from bregma under an angle of 10°), and mice were allowed to recover for 1 week after implantation.

For anterograde tracing experiments, recombinant AAVs expressing membrane-bound GFP (mGFP)/synaptophysin-mRuby were bilaterally injected into the LHb of *AADC^Cre* mice. After 4 weeks, mice were sacrificed for histological processing.

For optogenetic stimulation experiments, recombinant AAVs expressing ChR2, ChR2 with GFP, or ChR2 with sh-AADC were bilaterally injected into the LHb of *AADC^Cre* mice. After 3 weeks, the mice were humanely killed to obtain tissues for electrophysiology experiments.

For pharmacological experiments, recombinant AAVs expressing GFP or sh-AADC were bilaterally injected into the LHb of *VGAT^Cre*::Ai14

mice. After 3 weeks, the mice were humanely killed to obtain tissues for electrophysiology experiments.

For rescue experiments in the US model, recombinant AAVs expressing mAADC or EYFP were bilaterally injected into the LHb of *AADC^Cre* mice. After 19 days, the mice were exposed to learned helplessness-inducing US for 3 days before the start of behavioural testing.

## Behavioural assays

All behavioural assays were performed during the light phase except for the SPT, which was performed during the dark phase to maximise the consumption of the sucrose solution. In all behavioural experiments, the experimenter was blinded to the animals' genotypes and the experimental conditions; the data were also analysed in a blinded manner. Mice used for AADC knockdown and overexpression experiments were subjected to tail suspension stress after the completion of the behavioural tests and then were sacrificed.

## OFT

Each mouse was placed in a corner of the open-field chamber (45 cm × 45 cm × 40 cm) and allowed to explore for 15 min. The total distance moved, time spent in the centre, frequency of visits to the centre and latency to visit the centre were calculated using EthoVision XT 12 tracking software (Noldus Information Technology BV, Wageningen, the Netherlands).

## EZM

Mice were placed alone in a closed quadrant and allowed to explore the apparatus freely for 5 min. The apparatus consisted of two open quadrants and two closed quadrants and was elevated 60 cm above the floor. The total distance travelled, time spent in the open quadrants, frequency of visits to the closed quadrants and latency to visit the open quadrants were recorded by EthoVision XT 12 tracking software (Noldus Information Technology BV).

## TST

The TST was conducted using a four-chamber apparatus divided by opaque, matte-surfaced acrylic partitions. Each mouse was suspended by their tail in one of the chambers using adhesive tape. A video was recorded for 6 min, and the last 4 min was scored for the immobility time. Immobility was determined every 5 s, and the immobility time was calculated as the number of instances of immobility multiplied by 5 s. Any instance when the mouse did not move any of its limbs was scored as immobility.

## SPT

The SPT was conducted with modifications to a procedure previously described[62]. Singly housed mice were habituated to two water bottles containing tap water for one day and then were presented with two identical bottles, one with 1% sucrose solution and the other with tap water. To minimise the potential effect of side preference, the positions of the two bottles were switched daily. Sucrose and water consumption were recorded daily by reweighing the preweighed bottles of test solutions. Sucrose preference was calculated as a relative ratio as the mass of sucrose solution intake divided by the total fluid intake.

## Microdialysis

To examine extracellular levels of dopamine in vivo, a CMA 7 microdialysis probe with a 2-mm membrane length (P000083, CMA Microdialysis) was used. The probe was slowly inserted into the ventral striatum of mice through a guide cannula and connected to a single-channel liquid swivel and a counterbalancing system 2 h before the experiments. The probe was perfused at a flow rate of 1.0 μL pre min with a microinfusion pump with sterile artificial cerebrospinal fluid (ACSF; CMA Microdialysis) with the following composition (in mM):

147 NaCl, 2.7 KCl, 1.2 CaCl$_2$ and 0.85 MgCl$_2$. The perfusate was collected every 20 min with a refrigerated microfraction collector; the first three times, perfusate was collected for baseline measurements after 2 h of preperfusion, and then perfusate was collected six times after the injection of 5 mg per kg CNO in 0.5% DMSO and 1× PBS. Dopamine quantification was performed using an ultrahigh-performance liquid chromatography–tandem mass spectrometry system. This system consisted of an ExionLC system (SCIEX, Framingham, MA, USA) with an Acquity HSS T3 column (2.1 × 100 mm, 1.8 μm; Waters Corporation, Milford, MA, USA) and a quadrupole ion trap (QTRAP) 6500+ mass spectrometer (SCIEX) with an electrospray ionisation source. The data were acquired and quantified using Analyst software version 1.7 (SCIEX). The mobile phase consisted of 5 mM ammonium formate and 0.1% formic acid in water combined with 5 mM ammonium formate in acetonitrile:methanol 1:1 (v/v). The flow rate was 0.3 mL per min, and the injection volume was 10 μL. The mass spectrometer was optimised, and a multiple reaction monitoring scan was performed in positive ion mode. For histology, the experimental mice were anaesthetized with a mixture alfaxalone (40 mg per kg): xylazine (10 mg per kg) and perfused with 0.9% saline and then with 4% paraformaldehyde.

## Circuit tracing

To track mGFP/synaptophysin-mRuby neurons or terminals, mice were anaesthetized with a mixture alfaxalone (40 mg per kg): xylazine (10 mg per kg) and perfused with 0.9% saline and then 4% paraformaldehyde 4 weeks after virus injection. The brains of the mice were removed and postfixed in 4% paraformaldehyde overnight at 4 °C. Fixed brains were cut into 2-mm-thick midline sagittal sections that included the LHb and the RMTg using a sagittal brain matrix for mice (Harvard Apparatus, Holliston, MA, USA). Sagittal brain sections were optically cleared using a Tissue-Clearing Kit (HRTC-001 and SHMS-060, Binaree, Inc., Daegu, South Korea). Briefly, brain sections were incubated in the starting solution at 4 °C and then washed with distilled water three times for 1 h each time at 4 °C. The washed brain sections were incubated with the tissue-clearing solutions in a shaking incubator at 37 °C for 4 days and then incubated in the mounting and storage solution. A z-stack of the sections was acquired with a Lightsheet Z.1 fluorescence microscope (Carl Zeiss AG, Oberkochen, Germany), and the images were three-dimensionally reconstructed using Imaris software (Oxford Instruments, Abingdon, UK).

For the characterisation of presynaptic neurons in the LHb and postsynaptic neurons in the RMTg, FISH was performed using free-floating brain sections that included the LHb and the RMTg. For this analysis, the perfused and postfixed brains were equilibrated in RNase-free 30% sucrose and then cut into 40-μm-thick coronal sections on a cryotome (CM300, Leica). Floating brain sections were placed in chamber slides (Thermo Fisher Scientific) containing Pretreatment 2 (Advanced Cell Diagnostics) preheated to 60 °C and were then incubated for 10 min to allow digestion of sucrose cross-links. The brain sections were then pretreated with Pretreatment 4 (Advanced Cell Diagnostics) to allow protease digestion at room temperature for 30 min. Probe (GAD1) hybridisation, amplification and analysis were conducted as described above for FISH.

## Electrophysiology

For brain-slice preparation, mice were euthanised by isoflurane inhalation, and 300-μm-thick coronal brain slices containing the RMTg were obtained using a vibrating blade microtome (VT1000 S, Leica) in cold cutting solution with the following composition (in mM): 92 N-methyl-D-glucamine, 2.5 KCl, 1.25 NaH$_2$PO$_4$, 30 NaHCO$_3$, 30 HEPES, 0.5 CaCl$_2$, 10 MgCl$_2$ and 25 glucose, saturated with 95% O$_2$/5% CO$_2$. The slices were allowed to recover at room temperature for at least 1 h in a recovery solution with the following composition (in mM): 92 NaCl, 2.5 KCl, 1.25 NaH$_2$PO$_4$, 30 NaHCO$_3$, 30 HEPES, 2 CaCl$_2$, 2 MgCl$_2$ and 25 glucose, saturated with 95% O$_2$/5% CO$_2$. Each slice was transferred from

a recovery reservoir to the recording chamber of a fixed-stage upright microscope (BX51WI, Olympus, Tokyo, Japan) and perfused with oxygenated ACSF with the following composition (in mM): 124 NaCl, 2.5 KCl, 1.25 NaH$_2$PO$_4$, 24 NaHCO$_3$, 1.5 CaCl$_2$, 1.5 MgCl$_2$ and 10 glucose, saturated with 95% O$_2$/5% CO$_2$. ACSF was supplied to the chamber at 1.5–2 mL per min. Each submerged slice was visualised either directly via the microscope's optics or indirectly via an optiMOS high-resolution charge-coupled device camera system (QImaging Corporation, Surrey, Canada) attached to the microscope's video port.

Whole-cell patch-clamp recordings in current-clamp mode were used to measure spontaneous action-potential firing in RMTg GABAergic neurons indicated by red fluorescence in *VGAT*$^{Cre}$::Ai14 mice. The recordings were obtained using borosilicate glass pipettes (resistance 4–10 MΩ) prepared using a 2-stage vertical pipette puller (P-1000, Sutter Instrument, Novato, CA, USA; or PP-83, Narishige Scientific Instrument Lab, Tokyo, Japan). The pipettes were filled with a solution containing the following (in mM): 140 κ-gluconate, 10 HEPES, 0.5 EGTA, 10 glucose, 2 Na-ATP and 0.5 Na-GTP; the pH was adjusted to 7.2 with KOH. Signals were amplified and filtered (2 kHz) using a MultiClamp 700B amplifier, sampled at 5 kHz using a Digidata 1440 digitiser and recorded using pClamp 10 software (all from Molecular Devices, San Jose, CA, USA). No correction for liquid junction potential was made. Analyses were performed using Clampfit 10 (Molecular Devices). Only cells with stable access resistance of <25 MΩ were included in the analysis. RMPs were measured at baseline for 1 min after low-pass filtering (8-pole Bessel filter set at 0.5 Hz) and analysed within a 1-mV error range compared with unfiltered recordings.

To examine whether the RMTg directly receives synaptic input from LHb D-neurons, we investigated the responses of RMTg neurons using optogenetic stimulation in *AADC*$^{Cre}$ mice. For brief stimulation (10 ms), blue light (473 nm, 0.5~1 s and ~10 mW) was delivered by a light-emitting diode light source (PSU-III-LED, Changchun New Industries Optoelectronics Tech. Co., Ltd, Changchun, China) via an optic fibre during whole-cell recordings from RMTg neurons. Recording pipettes were filled with a solution containing the following (in mM): 127 CsMeSO$_4$, 10 NaCl, 5 EGTA, 4 Mg-ATP, 2 Na-GTP and 2 QX-314. To confirm the presence of glutamatergic inputs, optically evoked EPSCs (oEPSCs) were recorded during the application of the competitive AMPA/kainate receptor antagonist CNQX (20 µM; Bio-Techne Corporation, Minneapolis, MN, USA) and the selective NMDA receptor antagonist AP5 (50 µM; Bio-Techne Corporation).

For optogenetic stimulation experiments to confirm the function of LHb D-neurons using *AADC*$^{Cre}$ mice, blue light (473 nm, 10 mW, 60–80 pulses at 1 Hz) was delivered by a light-emitting diode light source via an optic fibre during whole-cell recordings from RMTg neurons.

For pharmacological experiments on RMTg GABAergic neurons using control mice and AADC-knockdown mice, action potentials were recorded under the effects of the TAAR1 agonist RO5263397 (500 nM; Bio-Techne Corporation) and the TAAR1 antagonist EPPTB (1 µM; Bio-Techne Corporation) in ACSF. All drugs were dissolved in 1% DMSO and stored at −70 °C before use. Recordings began at least 2 min after whole-cell recording was established via break-in. The frequency of action-potential firing was analysed based on a 30-s period after drug application. Following exposure to RO5263397 and EPPTB, the frequency and amplitude of spontaneous EPSCs (sEPSCs) were recorded with 10 µM bicuculline and 1 µM CGP 35348 in the extracellular solution to isolate excitatory currents and with the holding potential set at −70 mV. The recording pipette was filled with a caesium chloride-based internal solution.

## Drug administration

For in vivo drug administration, the selective TAAR1 agonist RO5263397 was dissolved in sterilised 1× PBS, and the selective TAAR1 antagonist EPPTB was dissolved in a solution containing 10% DMSO and 1% Tween 80 in sterilised 1× PBS. From 3 weeks after stereotactic surgery, the AADC-knockdown mice were intracerebroventricularly injected with saline or 1 µg per µL RO5263397 into the RMTg. From 7 days after cannula implantation, the naïve mice were intracerebroventricularly injected with vehicle (10% DMSO and 1% Tween 80 in sterilised 1× PBS) or 5 µg per µL EPPTB[63,64] into the RMTg. All injection steps were performed 30 min before the TST at 0.3 µL per min using a cannula connected through polyethene tubing to a Hamilton microsyringe attached to an UltraMicroPump III (World Precision Instruments). All injected mice were subjected to the TST for 6 min.

## In vivo recording of RMTg GABAergic neurons

*VGAT*$^{Cre}$ mice received bilateral infusions of recombinant AAVs expressing either GFP or sh-AADC in the LHb together with a unilateral injection of recombinant AAVs expressing ChR2 in the RMTg. Then an optrode array containing one optic fibre (200-µm diameter, 0.22 numerical aperture) and four tetrodes (20-µm diameter tungsten wire; California Fine Wire Co., Grover Beach, CA, USA) was implanted above the injected RMTg. After recovery, individual mice were placed in a holding cage, and their optrodes were connected to a diode-pumped, solid-state laser (Laserglow Technologies, Toronto, Canada) and a Digital Lynx recording system (Neuralynx, Boseman, MT, USA). Neural signals from the tetrodes were filtered between 0.6 to 6 kHz, digitised at 32 kHz and amplified 1000–8000 times. While unit spikes were monitored, 10 blue-light pulses (473 nm; 5-ms width at 30 Hz; 4–12 mW per mm$^2$ intensity) were delivered via the optic fibre. If no light-responsive units were detected, the optrodes were moved down in 40 to 80-µm increments, up to 160 µm per day. When at least two light-responsive units were observed on a given day, the behavioural recording session started on the next day. In each daily session, spontaneous firing patterns from RMTg neurons were recorded in the holding cage for the first 15 min, during tail suspension stress for the next 10 min and back in the cage for the final 20 min. At the end of the session, 10 light pulses were given 10 times to identify ChR2-expressing neurons in the RMTg. A total of 39 behavioural recording sessions were performed using five control and five AADC-knockdown mice (2–6 sessions per mouse).

To analyse the unit data, neuronal spikes were isolated based upon various waveform features using an Offline Sorter (Plexon Inc., Dallas, TX, USA). Only units showing stable firing throughout the behavioural recording session were further analysed using MATLAB software (MathWorks, Natick, MA, USA). To categorise GABAergic neurons, a cluster analysis was performed based on spike probability and latency in response to 100 light pulses. The cluster that showed a spike probability >0.81 and latency <5.1 ms was classified as GABAergic. These neurons also showed the highest correlations between spontaneous and light-evoked waveforms. To further examine the activity of GABAergic neurons before, during and after tail suspension stress, histograms (500-ms bins) were constructed during the three periods. Two-tailed Wilcoxon signed-rank tests were used to determine whether individual GABAergic neurons displayed significant changes in firing during the first 5-min period of tail suspension and during the 5-min period after the stress, relative to their basal firing rates measured for 5 min before the stress. Group comparisons of GABAergic firing were conducted using firing rates averaged across the entire GABAergic population.

## Statistical analysis

Samples were excluded from analyses if the viral injection sites were outside the LHb. Statistical analysis was conducted with IBM SPSS Statistics for Windows, version 26.0 (IBM Corp., Armonk, NY, USA). Statistical significance was determined using two-tailed unpaired *t* test or Mann–Whitney *U* tests for unpaired data, and two-tailed Wilcoxon signed-rank tests for paired data. For comparisons among more than two groups, one-way analysis of variance (ANOVA) was used with the appropriate post hoc tests. To monitor changes over time, one-way

repeated-measures ANOVA was run, followed by a contrast test. Data are expressed as the mean values ± standard error of the mean (s.e.m.), and indications of statistical significance correspond to the following values: $p^* < 0.05$, $p^{**} < 0.01$ and $p^{***} < 0.001$. Details of the statistical analyses, including the number of animals, the exact statistical tests used and the analysis results, are reported in Source Data and Supplementary Table 2.

### Reporting summary

Further information on research design is available in the Nature Portfolio Reporting Summary linked to this article.

## Data availability

The data that support the findings of this study are available from the corresponding author upon reasonable request. Source data are provided with this paper.

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

## Acknowledgements

We thank NeuroVIS for performing the in vivo measurements of dopamine release, and Ulsan National Institute of Science and Technology Optical Biomed Imaging Center (UOBC) for the three-dimensional analysis of the neural circuit. We thank Drs. Yongsang Cho (Gachon Liberal Arts College, Gachon University), Il Hwan Kim (University of Tennessee Health Science Center), Kihoon Han (College of Medicine, Korea University), Woong Sun (College of Medicine, Korea University) and Gi Hoon Son (College of Medicine, Korea University) for their critical reading and comments. This work was supported by the National Research Foundation of Korea (NRF), funded by the Ministry of Science, ICT and Future Planning (NRF-2017M3C7A1079692 and NRF-2020M3E5D9080794 to H.K.; NRF-2017R1D1A1B06032730 to H.W.L.; E0210201-01 to J.L.; NRF-2022M3E5E8017804 to Y.S.J.; and NRF-2018M3C7A1024597 to D.L.).

## Author contributions

S.H.Y., H.W.L. and H.K. designed the study. E.Y. performed the FISH analysis and histology experiments. J.L. and S.C. performed the in vitro patch-clamp experiments. H.Y. and H.S.P. produced the viruses and carried out the qPCR analysis. S.H.Y. performed the stereotactic surgery. S.H.Y., J.Y.K. and J.T.J. conducted the behavioural experiments and established the animal stress models. D.L. analysed the experimental data and drew the schematics. Y.S.J. and G.H.P. performed in vivo recording experiments. J.Y.P. produced the shRNA plasmid. H.W.L. and H.K. supervised all aspects of the work. S.H.Y., H.W.L. and H.K. wrote the manuscript with input from all authors.

## Competing interests

The authors declare no competing interests.
