## [Peer Review File · Nature Communications]

Neural mechanism of acute stress regulation by trace aminergic signalling in the lateral habenula in male miceREVIEWER COMMENTS

Reviewer #1 (Remarks to the Author):

The authors employed a comprehensive and complementary set of experimental approaches (viral knockdowns, chemogenetics, optogenetics, c-Fos, ephys, microdialysis, and behavioral assays) to examine how acute stress engages the LH-> RMTg -> VTA circuit. Their results illustrate that trace amine signaling from LH neurons provides inhibition onto GABAergic RMTg neurons, which in turn leads to greater activation of VTA dopamine neurons. These findings are very exciting and novel. There is one important experiment that is lacking (it is easily addressable). The authors could also provide a greater level of integration with prior studies in the discussion. That being said, this is an outstanding body of work.

Concerns:

- 1) One of the claims of the study is that trace aminergic signaling allows the brain to manage acute stress. However, the primary experiment that addresses this point is the tail suspension test that involved administering a TAAR1 agonist (RO) in mice with AADC knocked down (Fig. 5h). I can understand the need for performing this experiment in animals with AADC knocked down, as there may be a floor effect in control animals. However, by using an agonist one is not addressing whether trace aminergic signaling is having an endogenous role in mediating the effects of acute stress on behavior. To address the stated claim, the authors must instead administer the TAAR1 antagonist (EPPTB) into the RMTg in control animals. One would then expect a higher time immobile in these subjects relative to vehicle injected animals.
- 2) Why is the effect of RO (TAAR1 agonist) on raw firing rate smaller in the AADC knockdown animals? If anything I would expect a greater effect since there is a reduction in the presumed tonic levels of trace amines. This point should be discussed.
- 3) The discussion was quite limited. It is worth acknowledging that not all chronic stress protocols elicit a hypodopaminergic phenotype. A large body of evidence from the Han lab and others (see Chaudhury 2012 in particular) demonstrates that animals exhibiting depression-related behaviors (susceptible phenotype) following exposure to social defeat have higher baseline dopamine neuron firing rates. It could be possible that this effect may be mediated trace-amine signaling. Regardless, a greater synthesis of the current findings within the existing literature is warranted.
- 4) Any mention of 'depression' or 'depressive' should instead refer to 'depression-like' or 'depressive-like'. There are a number of instances in the manuscript where this occurred.

Typos:

Line 65: should be 'VTA TH-positive neurons' not 'VTA serotonin-positive neurons'.

Line 90: Please remove the reference to 'depression induced'. No behavioral assay was done with the qPCR analysis. The corresponding schematics in Fig. 3a,b should also remove 'depressed rat' and 'helpless mouse'.

Lines 131-132: Please rephrase. One does not measure firing frequency using optogenetics.

Reviewer #2 (Remarks to the Author):

This is a very well conducted study that identifies an important novel signalling mechanism in LHb D-neurons that allow them to provide inhibitory control over target neurons despite being glutamate-releasing. The experiments provide a clear mechanism for how stress-induced changes to intrinsic physiology of LHb D-neurons can guide opposing downstream effects on VTA neurons via the RMGTg in chronic vs acute exposures. The figures are well laid out, the data collection appears rigorous, and the logical flow of the experiments is strong. I particularly enjoyed the use of both mouse and rat models. I only have a few minor concerns:

1.The authors do not sufficiently setup the rationale and the background for this study in the introduction, nor do they provide a sufficient discussion of previous work in the conclusion section. Specifically, they should better introduce trace amines and what is known about their role in physiology and behavior. This may include identifying specific trace amines and how they are synthesized, what their signalling mechanisms are, and how engagement of the TAAR1 receptor impacts neurons and neural circuits. The exciting findings of this paper would be much more impactful if they were put into context better.

2.The proposed model is that chronic stress upregulates LHb activity and VTA DA neuron activity while acute stress upregulates LHb activity and downregulates VTA DA neuron activity. The majority of the paper investigates trace amine signalling as the mechanism driving the behavioral effects of acute stress; however, the qPCR experiments tested the effects of chronic stress on AADC expression. While the direction was as expected (decreased), I do think testing the effects of acute stress on AADC expression would be valuable here. It may be the case that AADC expression is not changed by acute stress but is still involved in the behavior, but I think this should be explicitly tested. Along the same lines, it may be helpful to also perform qPCR for TAAR1 in the RMTg after acute/chronic stress.

a.Related: At what time point post-stress was tissue collected for qPCR?

3.The authors do not provide appropriately zoomed-out images of viral infections for behavioral, tracing, and electrophysiological studies. As the LHb is such an anatomically small region (especially in mice), specific viral targeting is difficult. Thus, images of the viral infections verifying no spill-over into neighboring regions is needed (particularly the chemogenetic studies, the shRNA studies, and the circuit tracing studies). Of note, the shRNA viral infections shown in extended data figure 4 do not appear to be restricted to the LHb and extend well into the PVT and lateral to the LHb. Similarly, the images in Extended Data Fig 3 do not show the entire LHb or its boundaries, limiting assessment of the viral

targeting accuracy.

4. Page 4 line 65: says VTA serotonin neurons but I think the authors meant VTA dopamine neurons

5. Figure 3k/Extended data Fig 8: Was there a behavioral challenge for cFos studies or were the mice just sacrificed straight from their home cages?

Reviewer #3 (Remarks to the Author):

The authors provide evidence for paradoxical inhibition of the RMTg by LHb stimulation and acute stress via trace amine signaling. They also provide some evidence for decreased AADC expression underlying depression-related effects of chronic stress and inescapable shock. The results are surprising and would have a significant impact on the field. However, I have concerns about the methodology and their interpretations, many of which relate to the ambiguity of the timing of the effects they observe. I also have concerns about how their results fit with multiple published studies indicating a net excitatory effect of LHb stimulation on RMTg activity and inhibitory effect on dopamine neuron activity.

Concerns:

1. In Fig. 3k-n, it is unclear if the c-fos results were from animals exposed to acute stress or not. According to their model, knockdown of AADC in the LHb should reverse the changes in c-fos expression in the RMTg and DA neurons after acute stress (after AADC knockdown, acute stress should increase c-fos in RMTg and decrease it in DA neurons). It is not clear if this experiment was done. This would help mitigate the concern that their unexpected results are due to the insensitivity of c-fos assays to the timing of neuronal activation (see below).
2. Sanchez-Catalan et al, 2017 did not observe a decrease in c-fos in the RMTg with acute stressors, including restraint and forced swim test (<https://www.ncbi.nlm.nih.gov/pmc/articles/PMC5240171/>). It is not clear why these acute stressors, but not tail suspension (as used here), would fail to decrease c-fos in the RMTg, potentially limiting the generalizability of the authors' results.
3. Paul Shepard and others have shown that stimulation of the LHb decreases VTA DA activity and this is prevented by RMTg lesion (<https://www.ncbi.nlm.nih.gov/pmc/articles/PMC5214632/>). They also showed that lesions of LHb outputs block c-fos induction in the RMTg during acute footshock stress (<https://www.ncbi.nlm.nih.gov/pmc/articles/PMC3625179/>). How do the authors reconcile their data with these studies?
4. Tom Jhou's lab finds that acute stressors tend to activate RMTg neurons (<https://www.ncbi.nlm.nih.gov/pmc/articles/PMC6361585/>), so how do the authors reconcile this with their c-fos results? They also found that restraint stress and LiCl cause initial excitation followed by

rebound inhibition. Therefore, it seems possible that the authors are measuring an effect of delayed rebound inhibition during acute stress (or its termination) rather than immediate inhibition to acute stress.

5. It is unclear if chemogenetic stimulation increases the spiking of LHb neurons, which are normally very active in basal conditions (c-fos might go up even if there is depolarization block). Chemogenetic inhibition could be used to show that opposite effects are seen with neuronal inhibition, ruling out a depolarization block explanation for the effects of LHb stimulation.

6. The authors state that “little is known about how the activity of the RMTg changes under these (acute stress) circumstances”. But see <https://www.ncbi.nlm.nih.gov/pmc/articles/PMC6361585/>.

7. It is unclear what % of RMTg-projecting LHb neurons express AADC. This might help reconcile the authors’ results that indicate a net inhibitory interaction between the LHb and RMTg and others’ indicating a net excitatory interaction between the LHb and RMTg (e.g., <https://www.ncbi.nlm.nih.gov/pmc/articles/PMC5214632/>, <https://www.ncbi.nlm.nih.gov/pmc/articles/PMC3625179/>, see Supplementary Figure 1 in Stamatakis and Stuber, <https://www.ncbi.nlm.nih.gov/pmc/articles/PMC3411914/>). Related to this point and point #1 above, it is unclear if inhibitory trace amine signaling from the LHb can override glutamatergic transmission from ALL LHb efferents to the RMTg (not just ones from AADC+ terminals), as the authors seem to implicitly suggest. Does LHb activation during acute stress really inhibit the RMTg? I think more than c-fos is required to answer this question.

8. It is not clear if chronic restraint stress and learned helplessness caused depression-related phenotypes and if this can be rescued by restoring AADC expression in the LHb.

9. AADC knockdown increased RMTg firing in slices but did not change the resting membrane potential. So how was firing rate changed? This seems to contradict Bradaia et al., 2009, PNAS (<https://pubmed.ncbi.nlm.nih.gov/19892733/>) who showed that TAAR1 activates a potassium channel to change DA neuron firing.

10. The authors optogenetically stimulate D-neuron terminals and find a decrease in RMTg firing (fig 5b), but the timing of this effect is unclear. It would be better if the authors show firing rate in a continuous plot that includes before optogenetic stimulation and after stimulation to see the timecourse of the inhibition, especially since this effect is contrary to published results for LHb-RMTg stimulation (see Supplementary Figure 1 in Stamatakis and Stuber, <https://www.ncbi.nlm.nih.gov/pmc/articles/PMC3411914/>).

11. It is unclear what stimulation protocol the authors used for their optogenetic stimulation of LHb terminals in slices. They state that they used 0.5 ~ 1 second illumination, but then later say blue light was applied between 60-80 sec. If the authors used 20 seconds of constant stimulation, this is problematic as it would be expected to deplete neurotransmitters from LHb terminals over time.

12. The authors find that D-neuron terminal stimulation decreases RMTg firing in slices, but oddly, do not show that the TAAR1 receptor antagonist blocks this effect. This would be superior to relying on (long-term) AADC knockdown to block the effect and would show that the inhibition is due to TAAR1 activation.

13. Typo on Line 65, says "serotonin-positive" (should be "TH-positive"?).

December 16, 2022

Re: **Neural mechanism of acute stress management by trace aminergic signalling in the lateral habenula**

Reviewer #1 (Remarks to the Author):

The authors employed a comprehensive and complementary set of experimental approaches (viral knockdowns, chemogenetics, optogenetics, c-Fos, ephys, microdialysis, and behavioural assays) to examine how acute stress engages the LH-> RMTg -> VTA circuit. Their results illustrate that trace amine signalling from LH neurons provides inhibition onto GABAergic RMTg neurons, which in turn leads to greater activation of VTA dopamine neurons. These findings are very exciting and novel. There is one important experiment that is lacking (it is easily addressable). The authors could also provide a greater level of integration with prior studies in the discussion. That being said, this is an outstanding body of work.

1) One of the claims of the study is that trace aminergic signalling allows the brain to manage acute stress. However, the primary experiment that addresses this point is the tail suspension test that involved administering a TAAR1 agonist (RO) in mice with AADC knocked down (Fig. 5h). I can understand the need for performing this experiment in animals with AADC knocked down, as there may be a floor effect in control animals. However, by using an agonist one is not addressing whether trace aminergic signalling is having an endogenous role in mediating the effects of acute stress on behaviour. To address the stated claim, the authors must instead administer the TAAR1 antagonist (EPPTB) into the RMTg in control animals. One would then expect a higher time immobile in these subjects relative to vehicle injected animals.

Response: We thank the reviewer for the valuable suggestion. As the reviewer suggests, we conducted a tail suspension test (TST) following the EPPTB treatment. Administration of EPPTB (5 µg/µL) to the RMTg of naïve mice significantly increased immobility time in the TST compared with the vehicle-injected mice. These additional data strongly support our conclusion that trace amines produced by LHb D-neurons can sufficiently modulate RMTg GABAergic neurons in response to acute stress. We added this data to **Supplementary Fig. 15** and described the result in the revised manuscript (**lines 202-208**).

2) Why is the effect of RO (TAAR1 agonist) on raw firing rate smaller in the AADC knockdown animals? If anything I would expect a greater effect since there is a reduction in the presumed tonic levels of trace amines. This point should be discussed.

Response: We appreciate the reviewer's keen advice. First, we examined the mRNA expression level of TAAR1 to rule out a possible downregulation of its expression in the AADC-knockdown and control mice. However, there was no significant difference between control and AADC-knockdown mice (**Supplementary Fig. 13a and b**). Moreover, we could not find any TAAR1 alterations in the RMTg from mice that underwent acute stress or US

(Supplementary Fig. 13c-f).

Given this invariable nature of RMTg TAAR1 expression, the diminished RMTg responsiveness to RO5263397 in AADC-knockdown mice may not be caused by the altered level of TAAR1 in RMTg, but could be a consequence of an unknown effect originating from long-term AADC knockdown in the LHb that could change the properties of cells in the RMTg, as evidenced by our electrophysiology results (**Fig. 4g**). Thus, further follow-up studies are needed to provide a clearer answer to this question.

We included the new data in the *Results* and *Discussion* (**lines 183-194 and 303-306**).

3) The discussion was quite limited. It is worth acknowledging that not all chronic stress protocols elicit a hypodopaminergic phenotype. A large body of evidence from the Han lab and others (see Chaudhury 2012 in particular) demonstrates that animals exhibiting depression-related behaviours (susceptible phenotype) following exposure to social defeat have higher baseline dopamine neuron firing rates. It could be possible that this effect may be mediated trace-amine signalling. Regardless, a greater synthesis of the current findings within the existing literature is warranted.

Response: We thank the reviewer for the constructive comment. Following the reviewer's suggestion, we tested whether chronic social defeat stress (CSDS) can affect AADC expression in the LHb. For this, we employed a CSDS protocol used in a previous paper (Biological Psychiatry. 2022. 92, 104–115) to make CSDS-susceptible mice. For CSDS, each eight-week-old male mouse was exposed to 10 min of physical aggression by a 3-month-old male ICR (CD-1) mouse. After the session, the defeated mice were housed overnight within the same cage as the ICR mice on the opposite side of a transparent and perforated divider to provide sensory, but not physical, contact. The procedure was repeated for 10 consecutive days with a new aggressor on each day. All stressed mice received the social interaction test 24 hours after the last defeat episodes and the sucrose preference test to assess depressive-like phenotypes. We then carried out a fluorescent in situ hybridisation (FISH) analysis to measure AADC mRNA in the LHb. However, we found no difference in AADC expression in the LHb of CSDS-susceptible mice compared with control mice (**see figure below**; $**p < 0.01$ and $***p < 0.001$; ns, not significant; this figure is NOT included in the revised manuscript).

As CSDS activates LHb and lateral hypothalamus (LH) neurons (afferent neurons of the LHb) (Neurobiol Stress. 2021. 14, 100298) and as LHb D-neurons lowered the activity of RMTg GABAergic neurons (our results), we expected that continuous activation of VTA dopaminergic neurons could be maintained by RMTg inhibition through trace aminergic signalling of D-neurons. Indeed, activation of the LH-LHb circuit alleviates social avoidance and despair-like and anxiety-like behaviours after CSDS (Neurobiol Stress. 2021. 14, 100298). It will be very interesting to study the role of LHb D-neurons in the mesolimbic system via the RMTg under CSDS. We described this possibility in detail in the *Discussion* (**lines 286-302**).

4) Any mention of ‘depression’ or ‘depressive’ should instead refer to ‘depression-like’ or ‘depressive-like’. There are a number of instances in the manuscript where this occurred.

Response: We agree with the reviewer’s comment. We have replaced the terms ‘depression’ and ‘depressive’ with ‘depression-like’ and ‘depressive-like’ throughout the manuscript.

Typos:

Line 65: should be ‘VTA TH-positive neurons’ not ‘VTA serotonin-positive neurons’.

Response: We apologise for this obvious mistake. We corrected ‘VTA serotonin-positive neurons’ to ‘VTA TH-positive neurons’ (**line 78**).

Line 90: Please remove the reference to ‘depression induced’. No behavioural assay was done with the qPCR analysis. The corresponding schematics in Fig. 3a, b should also remove ‘depressed rat’ and ‘helpless mouse’.

Response: We deleted the reference and changed ‘depression induced by chronic restraint stress (CRS) or learned helplessness (LH) protocols’ to ‘exposed to chronic restraint stress (CRS) or learned helplessness-inducing unpredictable electric foot shocks (US)’ (**lines 103-105**). In **Fig. 3a and b**, we changed ‘depressed rat’ and ‘helpless mouse’ to ‘rat’ and ‘mouse’.

Lines 131-132: Please rephrase. One does not measure firing frequency using optogenetics.

Response: We replaced ‘using optogenetics’ with ‘using whole-cell patch-clamp recordings with optogenetic stimulation’ (**lines 156-157**).

Reviewer #2 (Remarks to the Author):

This is a very well conducted study that identifies an important novel signalling mechanism in Lhb D-neurons that allow them to provide inhibitory control over target neurons despite being glutamate-releasing. The experiments provide a clear mechanism for how stress-induced changes to intrinsic physiology of Lhb D-neurons can guide opposing downstream effects on VTA neurons via the RMTg in chronic vs acute exposures. The figures are well laid out, the data collection appears rigorous, and the logical flow of the experiments is strong. I particularly enjoyed the use of both mouse and rat models. I only have a few minor concerns:

1. The authors do not sufficiently setup the rationale and the background for this study in the introduction, nor do they provide a sufficient discussion of previous work in the conclusion section. Specifically, they should better introduce trace amines and what is known about their role in physiology and behaviour. This may include identifying specific trace amines and how they are synthesized, what their signalling mechanisms are, and how engagement of the TAAR1 receptor impacts neurons and neural circuits. The exciting findings of this paper would be much more impactful if they were put into context better.

Response: We thank the reviewer for the constructive comment. We provided background information on trace amines in the *Introduction* (lines 35-48) and expounded on trace amine-TAAR1 receptor signalling in the *Discussion* (lines 263-285).

2. The proposed model is that chronic stress upregulates Lhb activity and VTA DA neuron activity while acute stress upregulates Lhb activity and downregulates VTA DA neuron activity. The majority of the paper investigates trace amine signalling as the mechanism driving the behavioural effects of acute stress; however, the qPCR experiments tested the effects of chronic stress on AADC expression. While the direction was as expected (decreased), I do think testing the effects of acute stress on AADC expression would be valuable here. It may be the case that AADC expression is not changed by acute stress but is still involved in the behaviour, but I think this should be explicitly tested. Along the same lines, it may be helpful to also perform qPCR for TAAR1 in the RMTg after acute/chronic stress.

a. Related: At what time point post-stress was tissue collected for qPCR?

Response: We acknowledge that qPCR will be helpful; however, it is difficult to dissect out the RMTg because of its irregularly shaped structure. Moreover, it is difficult to avoid contamination by nearby tissues when collecting such small amounts of tissue. Furthermore, TAAR1 expression can only be measured in GABAergic neurons in the RMTg. Thus, we utilised FISH approach to achieve more precise measurements in these challenging areas. When we performed FISH analysis of TAAR1 in the RMTg after acute or chronic stress, we also observed the expression of AADC mRNA in the Lhb. As you know, we could analyse mRNA expression with FISH at the regional and cellular level using HALO v2.3.2089.18 software (Indica Labs). We also performed qPCR on mRNA from the Lhb after tail suspension stress. AADC expression in the Lhb was not different between tail suspension stress-exposed and naïve mice (see figure below; this figure is NOT included in the revised manuscript).

We measured changes in AADC and TAAR1 mRNA expression using FISH by counting the mean number of AADC or TAAR1 copies per μm^2 in the Lhb and the RMTg of the three groups of mice exposed to three acute stressors: 1) tail suspension for 10 min; 2) forced swimming for 6 min; 3) restraint stress for 6 min. All the mice were sacrificed 25 min after the

onset of acute stress, and samples were collected for FISH analysis (see **Methods; lines 373-378**). The three acute stressors did not alter AADC or TAAR1 expression levels in the LHb or the RMTg, respectively (**Supplementary Fig. 3a, 3f, 13c and 13d**). In addition, we determined the change of TAAR1 expression in the RMTg of the mice exposed to unpredictable electric foot shocks (see **Methods; lines 383-386**) and of the AADC-knockdown mice. Only chronic stress decreased AADC expression in the LHb without changing TAAR1 expression in the RMTg. We added this data as **Supplementary Fig. 13a, b, e and f** and described the new data in the **Results (lines 183-194)**.

3. The authors do not provide appropriately zoomed-out images of viral infections for behavioral, tracing, and electrophysiological studies. As the LHb is such an anatomically small region (especially in mice), specific viral targeting is difficult. Thus, images of the viral infections verifying no spill-over into neighbouring regions is needed (particularly the chemogenetic studies, the shRNA studies, and the circuit tracing studies). Of note, the shRNA viral infections shown in extended data figure 4 do not appear to be restricted to the LHb and extend well into the PVT and lateral to the LHb. Similarly, the images in Extended Data Fig 3 do not show the entire LHb or its boundaries, limiting assessment of the viral targeting accuracy.

Response: We added the zoomed-out images of viral infections to the chemogenetic studies (**Fig. 2a**), the shRNA studies (**Supplementary Fig. 6e**), the optogenetic studies (**Fig. 4d**) and the circuit tracing studies (**Supplementary Fig. 4b**). Given the difficult nature of specific viral targeting in the LHb, we instead showed the limited expression of AADC mRNA solely observed in the LHb (Allen Brain Atlas and our FISH data; **Supplementary Fig. 6a and b**). We believe the expansion of AAV into the PVT and lateral to the LHb poses no obstacle to the analysis of our data.

4. Page 4 line 65: says VTA serotonin neurons but I think the authors meant VTA dopamine neurons.

Response: We apologise for this obvious mistake. We corrected 'VTA serotonin-positive neurons' to 'VTA TH-positive neurons' (**line 78**).

5. Figure 3k/Extended data Fig 8: Was there a behavioural challenge for cFos studies or were the mice just sacrificed straight from their home cages?

Response: We obtained brain samples after tail suspension stress and generated FISH data for **Fig. 3k**. We added ‘sacrifice’ to the schematic drawing of **Fig. 3f**. We used naïve mice to measure normal gene expression patterns as shown in **Supplementary Fig. 12**.

Reviewer #3 (Remarks to the Author):

The authors provide evidence for paradoxical inhibition of the RMTg by LHb stimulation and acute stress via trace amine signaling. They also provide some evidence for decreased AADC expression underlying depression-related effects of chronic stress and inescapable shock. The results are surprising and would have a significant impact on the field. However, I have concerns about the methodology and their interpretations, many of which relate to the ambiguity of the timing of the effects they observe. I also have concerns about how their results fit with multiple published studies indicating a net excitatory effect of LHb stimulation on RMTg activity and inhibitory effect on dopamine neuron activity.

1. In Fig. 3k-n, it is unclear if the c-fos results were from animals exposed to acute stress or not. According to their model, knockdown of AADC in the LHb should reverse the changes in c-fos expression in the RMTg and DA neurons after acute stress (after AADC knockdown, acute stress should increase c-fos in RMTg and decrease it in DA neurons). It is not clear if this experiment was done. This would help mitigate the concern that their unexpected results are due to the insensitivity of c-fos assays to the timing of neuronal activation (see below).

Response: We apologise for the lack of clarity regarding this procedure. The data for **Fig. 3k-n** were collected from mice exposed to tail suspension stress prior to sacrifice. We added ‘sacrifice’ to **Fig. 3f**. and described the detailed procedure in the *Methods* (lines 489-491).

2. Sanchez-Catalan et al, 2017 did not observe a decrease in c-fos in the RMTg with acute stressors, including restraint and forced swim test (<https://www.ncbi.nlm.nih.gov/pmc/articles/PMC5240171/>). It is not clear why these acute stressors, but not tail suspension (as used here), would fail to decrease c-fos in the RMTg, potentially limiting the generalizability of the authors’ results.

Response: We thank the reviewer for the valuable comment. To generalise our results, we conducted comprehensive experiments as follows.

To confirm the reduction of c-Fos in the RMTg by acute stress, we employed other acute stressors – ‘forced swimming’, ‘restraint stress’ and ‘tail suspension stress’ – and carefully analysed c-Fos expression in the LHb and the RMTg. As shown in newly added **Supplementary Fig. 3**, all the acute stress conditions increased c-Fos mRNA expression in LHb D-neurons while decreasing it in RMTg GABAergic neurons. We believe these consistent results from multiple acute stress conditions can support our conclusion. We also believe our sensitive FISH approach can capture the subtle but significant differences in mRNA quantity at a cellular resolution.

3. Paul Shepard and others have shown that stimulation of the LHb decreases VTA DA activity and this is prevented by RMTg lesion (<https://www.ncbi.nlm.nih.gov/pmc/articles/PMC5214632/>). They also showed that lesions of LHb outputs block c-fos induction in the RMTg during acute footshock stress (<https://www.ncbi.nlm.nih.gov/pmc/articles/PMC3625179/>). How do the authors reconcile their data with these studies?

Response: In accordance with the reviewer's comment #4, we carried out in vivo recordings in the RMTg during acute stress. Inhibition of the activity of VTA dopaminergic neurons for a short period of time (within 1 sec) is thought to be caused by glutamatergic input of the LHb on the RMTg, which is consistent with our results (**Fig. 4e**).

One of the major discoveries in our study is the biphasic effect: the LHb initially excites the RMTg via glutamatergic signalling and subsequently rebound-inhibits it via trace aminergic signalling (**Fig. 6c-e**). Based on this model, the attenuation of inhibition of a dopaminergic neuron's activity following an RMTg lesion can be explained simply as a consequence of the reduction of RMTg GABAergic neurons to the response to LHb stimulation.

Furthermore, because LHb lesions can deprive both glutamatergic and trace aminergic inputs to the RMTg, this process can block acute c-Fos induction in the RMTg primarily via the loss of glutamates secreted in the LHb during acute footshock stress.

4. Tom Zhou's lab finds that acute stressors tend to activate RMTg neurons (<https://www.ncbi.nlm.nih.gov/pmc/articles/PMC6361585/>), so how do the authors reconcile this with their c-fos results? They also found that restraint stress and LiCl cause initial excitation followed by rebound inhibition. Therefore, it seems possible that the authors are measuring an effect of delayed rebound inhibition during acute stress (or its termination) rather than immediate inhibition to acute stress.

Response: We are grateful to the reviewer for the constructive comment, which greatly improved the quality of our manuscript. As described in the *Response to Comment #3*, we performed in vivo recording on the RMTg for 30 min under tail suspension stress with control mice and LHb AADC-knockdown mice. As shown in **Fig. 4f**, the basal firing rates in RMTg GABAergic neurons were increased by AADC knockdown in the LHb (**Supplementary Fig. 16g**). In control mice, tail suspension stress caused initial excitation followed by rebound inhibition (**Fig. 6c-e**). However, in AADC-knockdown mice, tail suspension stress elicited initial excitation but not rebound inhibition (**Fig. 6f-h**). Therefore, these new data are consistent with the reviewer's speculation that trace amines secreted from the LHb cause an effect of delayed rebound inhibition during acute stress (or its termination) rather than immediate inhibition to acute stress. We described these results in the *Results* and *Discussion* (**lines 210-236 and 248-262**).

5. It is unclear if chemogenetic stimulation increases the spiking of LHb neurons, which are normally very active in basal conditions (c-fos might go up even if there is depolarization block). Chemogenetic inhibition could be used to show that opposite effects are seen with neuronal inhibition, ruling out a depolarization block explanation for the effects of LHb stimulation.

Response: As suggested by the reviewer, we performed chemogenetic inhibition using hM4Di DREADD (pAAV-hSyn-DIO-hM4Di-mCherry; 44362, Addgene). After conducting the tail suspension test for 10 min, LHb D-neurons from control virus-injected mice (pAAV-hSyn-DIO-mCherry) expressed c-Fos, but DREADDi virus-injected mice did not (see figure below; this result is NOT included in the revised manuscript).

6. The authors state that “little is known about how the activity of the RMTg changes under these (acute stress) circumstances”. But see <https://www.ncbi.nlm.nih.gov/pmc/articles/PMC6361585/>.

Response: We removed ‘little is known about how the activity of the RMTg changes under these circumstances’ from our manuscript and revised the section (lines 50-60).

7. It is unclear what % of RMTg-projecting LHb neurons express AADC. This might help reconcile the authors’ results that indicate a net inhibitory interaction between the LHb and RMTg and others’ indicating a net excitatory interaction between the LHb and RMTg (e.g., <https://www.ncbi.nlm.nih.gov/pmc/articles/PMC5214632/>, <https://www.ncbi.nlm.nih.gov/pmc/articles/PMC3625179/>, see Supplementary Figure 1 in Stamatakis and Stuber, <https://www.ncbi.nlm.nih.gov/pmc/articles/PMC3411914/>). Related to this point and point #1 above, it is unclear if inhibitory trace amine signaling from the LHb can override glutamatergic transmission from ALL LHb efferents to the RMTg (not just ones from AADC+ terminals), as the authors seem to implicitly suggest. Does LHb activation during acute stress really inhibit the RMTg? I think more than c-fos is required to answer this question.

Response: We appreciate your comments and acknowledge that quantifying the proportion of neurons that express AADC among RMTg-projected LHb neurons would enable us to better understand our results in context with others’ results. Since the majority of RMTg-projecting LHb neurons are glutamatergic neurons, we attempted to measure the proportion of AADC-expressing neurons among RMTg-projecting LHb glutamatergic neurons. As a first step, we made retrograde AAVs (Cre-dependent rAAV-EF1 α -double floxed-EYFP) and injected them into the RMTg in *VGLUT2^{Cre}* mice. Then, we tried to identify the AADC-expressing neurons among retrogradely traced glutamatergic neurons in LHb using FISH and immunohistochemistry (IHC). However, we could not clearly label LHb AADC neurons due

to the failure to optimize the condition for FISH in paraformaldehyde perfused LHb sections and inadequate quality of available antibodies (10166-1-AP, Proteintech; NBP1-56918, Novus Biologicals) for IHC in the LHb. To overcome these problems, we performed in vivo recording of RMTg GABAergic neurons during tail suspension stress (**Fig. 6a**).

We believe that our in vivo recording results will help address the reviewer's comments, because the secretion of trace amines occurs by membrane diffusion and not by synaptic vesicles, and because TAAR1 is a GPCR with a slower response time than NMDA and AMPA. We found that among the Chr2-expressing GABAergic neurons in the RMTg during tail suspension stress, the percentage of neurons recorded with initial excitation was 94.1% (32/34) and with rebound inhibition was 91.2% (31/34) in control mice (**Fig. 6c and Supplementary Fig. 17a and b**). These percentages were higher than those of the AADC/VGLUT2 double-positive neurons (**Supplementary Fig. 1j**). This unbiased experimental result indicates that trace amines secreted from the LHb can sufficiently regulate the activity of all RMTg GABAergic neurons. Thus, we suggest that LHb D-neurons can subsequently override glutamate-mediated excitation, leading to the decreased activity of RMTg GABAergic neurons (rebound inhibition). Interestingly, LHb AADC knockdown gave a lower magnitude of excitation during the first 5-min period of tail suspension compared with the control (**Supplementary Fig. 17a and c**). It seems that the long-term effects of AADC knockdown might alter cellular responsiveness. Further studies are needed to provide a clearer answer to this suggestion. We added these results to the revised manuscript (**lines 210-236**).

8. It is not clear if chronic restraint stress and learned helplessness caused depression-related phenotypes and if this can be rescued by restoring AADC expression in the LHb.

Response: We previously reported that a chronic restraint stress rat model and a learned helplessness mouse model exhibited depression-related phenotypes (Scientific Reports. 2017. 7, 900.; Behav Brain Res. 2017. 322, 138–144). In accordance with the reviewer's comment, we replaced 'depression induced by chronic restraint stress (CRS) or learned helplessness (LH) protocols' with 'exposed to chronic restraint stress (CRS) or learned helplessness-inducing unpredictable electric foot shocks (US)' (**lines 103-105**) and 'depressed rat' and 'helpless mouse' with 'rat' and 'mouse' (**Fig. 3a and b**).

To investigate whether restoring AADC expression can rescue depression-related phenotypes caused by US, we constructed a mouse AADC expression vector (Cre-dependent pAAV-EF1 α -double floxed-AADC-EYFP) and generated AADC-overexpressing mice using AADC^{Cre} transgenic mice (**Supplementary Figure 9a-c**). AADC-overexpressing mice rescued despair-like and anhedonia-like behaviours after the TST and SPT, respectively, with no anxiety level changes after the OFT and EZM (**Supplementary Fig. 9d-m**). We added these results to the revised manuscript (**lines 122-135**).

9. AADC knockdown increased RMTg firing in slices but did not change the resting membrane potential. So how was firing rate changed? This seems to contradict Bradaia et al., 2009, PNAS (<https://pubmed.ncbi.nlm.nih.gov/19892733/>) who showed that TAAR1 **activates** a potassium channel to change DA neuron firing.

Response: We are indebted to the reviewer for pointing out important differences from a previously published manuscript. We recalculated substantial resting membrane potentials

(RMPs) from our previously recorded neurons. Indeed, the spontaneous and continuous firing of RMTg GABAergic neurons made it difficult to measure the actual resting membrane potentials. We filtered the action potentials with a low-pass filter (8-pole Bessel filter set at 0.5 Hz) during the 1-min recordings and recalculated the RMPs. The RMPs were analysed within a 1-mV error range compared with unfiltered recordings. As a result, unlike previous data, we confirmed that RMPs of RMTg GABAergic neurons were meaningfully increased in AADC-knockdown mice compared with control mice (**Fig. 4g**), consistent with the previously reported analysis in dopaminergic neurons (PNAS. 2009. 24, 106). These new results indicate that increased firing in RMTg GABAergic neurons from AADC-knockdown mice may be caused by the same mechanism as in dopaminergic neurons. These results are reflected in the revised manuscript (**lines 148-151 and 586-588**).

10. The authors optogenetically stimulate D-neuron terminals and find a decrease in RMTg firing (fig 5b), but the timing of this effect is unclear. It would be better if the authors show firing rate in a continuous plot that includes before optogenetic stimulation and after stimulation to see the time course of the inhibition, especially since this effect is contrary to published results for LHb-RMTg stimulation (see Supplementary Figure 1 in Stamatakis and Stuber, <https://www.ncbi.nlm.nih.gov/pmc/articles/PMC3411914/>).

Response: We apologise for the lack of full traces for the optogenetic studies. We added the time course of optogenetic stimulation as **Supplementary Fig. 11a**. Our results showed a decreased firing frequency of RMTg neurons immediately after optogenetic stimulation of LHb D-neuron terminals.

We demonstrated that stimulation of LHb D-neuron terminals elicited oEPSCs in RMTg neurons, which were blocked by CNQX/AP5 (**Fig. 4d and e**). To evaluate firing frequency during stress over the time course, we conducted an additional in vivo recording. The result showed that rebound inhibition of RMTg GABAergic neurons following stress is most likely caused by the trace amines produced by LHb D-neurons; the rebound inhibition was not observed in AADC-knockdown mice (**Fig. 6a-h**). Therefore, we believe that the EPSC is generated in RMTg neurons through glutamatergic signalling by LHb D-neurons, which induces decreased firing frequency in RMTg neurons by subsequent trace aminergic signalling.

11. It is unclear what stimulation protocol the authors used for their optogenetic stimulation of LHb terminals in slices. They state that they used 0.5 ~ 1 second illumination, but then later say blue light was applied between 60-80 sec. If the authors used 20 seconds of constant stimulation, this is problematic as it would be expected to deplete neurotransmitters from LHb terminals over time.

Response: We apologise for the confusion. For light stimulation, light pulses were 473 nm, 10 mW and 10 ms. To determine any changes in RMTg GABAergic neuronal firing in AADC^{Cre} mice, optical stimulation was delivered by 60 to 80 pulses at 1 Hz. We corrected this in the *Methods* (**lines 599-601**).

12. The authors find that D-neuron terminal stimulation decreases RMTg firing in slices, but oddly, do not show that the TAAR1 receptor antagonist blocks this effect. This would be

superior to relying on (long-term) AADC knockdown to block the effect and would show that the inhibition is due to TAAR1 activation.

Response: We are grateful to the reviewer for the constructive comment. We measured the firing frequency while administering EPPTB during optogenetic stimulation of LHb D-neuron terminals in the RMTg. We found that the decreased RMTg firing frequency by optogenetic stimulation was reversed by EPPTB treatment into the RMTg. These additional data strongly support our conclusion that inhibition of RMTg activity by LHb D-neurons is caused by trace amine-TAAR1 signalling. We added these data to **Supplementary Fig. 14** and described the result in the revised manuscript (**lines 195-198**).

13. Typo on Line 65 says “serotonin-positive” (should be “TH-positive”?).

Response: We apologise for this obvious mistake. We corrected ‘VTA serotonin-positive neurons’ to ‘VTA TH-positive neurons’ (**line 78**).

REVIEWER COMMENTS

Reviewer #1 (Remarks to the Author):

The authors were very responsive to the previous round of review. They addressed all my prior concerns. I have nothing further to add. Great work.

Reviewer #2 (Remarks to the Author):

The authors have much improved the manuscript with their revisions and have addressed my technical concerns. The manuscript is truly a wonderful body of work and will have important implications for the field.

I have one additional concern-- At the very least, the authors should provide a statement in their manuscript detailing the rationale for why they did not include any female animals in their study. Considering the fact that these stress models are easily performed in females, it seems like a large oversight to not include them.

If the authors cannot provide sufficient rationale, the fact that females were not included in this study should be mentioned as a major weakness. It would be great if the authors could recapitulate at least one of their major findings in females, but considering the large amount of data in the current manuscript it would be acceptable to investigate this in a subsequent paper.

Reviewer #3 (Remarks to the Author):

The authors add important new data showing that RMTg activity is initially increased during tail suspension, then decreased and AACD knockdown in the LHb reduces these effects. These data help resolve some, but not all, of the conflict between their data indicating a net inhibitory effect of LHb-RMTg stimulation, and others that have shown a net excitatory effect.

Concerns:

1. The concern that the results of HM3D activation might be due to depolarization block of spiking in the LHb was not addressed by their new HM4D experiment, which only examined c-fos in the LHb after tail suspension. Showing that activation of HM4D in the LHb (which should decrease LHb activity) increases c-fos in the RMTg after tail suspension (compared to controls with tail suspension) would address the concern.
2. The authors did not show that the TAAR1 antagonist blocked the effect of optogenetic LHb-RMTg

stimulation with their new experiment, but rather that the TAAR1 antagonist can reverse the effect of optogenetic LHb-RMTg stimulation (Supplementary Fig. 14) – so potentially, the TAAR1 antagonist is not blocking the effect of LHb-RMTg stimulation but just has an opposite effect on spiking. To show that the antagonist blocks the effect of LHb-RMTg stimulation, they would need to add the antagonist minutes before they start the optogenetic stimulation and show that there is no decrease in spiking during optogenetic stimulation with the antagonist present compared to the baseline of antagonist alone (w/o optogenetic stimulation). This experiment would not only show that TAAR1 activation causes the decrease in RMTg spiking, but would also show that optogenetic stimulation of LHb-RMTg increases the activation of TAAR1. I think this is important because, as the authors note, others found that depolarization does not increase the release of trace amines, so it is unclear if/how LHb activation would release trace amines to activate TAAR1 in the RMTg – a central component of their model. The authors allude to a “slow increase in the electrochemical gradient” (lines 273-4), but it is unclear what electrochemical gradient they are referring to, and I see no evidence to support this speculation either in their paper or in their citations.

3. The authors should acknowledge and discuss results from other studies showing that c-fos increases in the RMTg after acute footshock stress (Jhou et al, <https://www.ncbi.nlm.nih.gov/pmc/articles/PMC2841475/>; Brown and Shepard, <https://www.ncbi.nlm.nih.gov/pmc/articles/PMC3625179/>). Perhaps the authors can reconcile these results with their own, which found decreases in c-fos in the RMTg after other forms of acute stress.

March 15, 2023

Re: **Neural mechanism of acute stress management by trace aminergic signalling in the lateral habenula in male mice**

Reviewer #1 (Remarks to the Author):

The authors were very responsive to the previous round of review. They addressed all my prior concerns. I have nothing further to add. Great work.

Reviewer #2 (Remarks to the Author):

The authors have much improved the manuscript with their revisions and have addressed my technical concerns. The manuscript is truly a wonderful body of work and will have important implications for the field.

I have one additional concern-- At the very least, the authors should provide a statement in their manuscript detailing the rationale for why they did not include any female animals in their study. Considering the fact that these stress models are easily performed in females, it seems like a large oversight to not include them.

If the authors cannot provide sufficient rationale, the fact that females were not included in this study should be mentioned as a major weakness. It would be great if the authors could recapitulate at least one of their major findings in females, but considering the large amount of data in the current manuscript it would be acceptable to investigate this in a subsequent paper.

Response: We would like to express our sincere gratitude for reviewing our manuscript and providing valuable feedback on our revisions. We appreciate your insightful comments and would like to address your concern regarding the absence of female animals in our study.

We recognize that including female animals in our study would have been ideal, and we understand the importance of considering both sexes in preclinical research. However, due to the unpredictable impact of the estrous cycle on stress response, anxiety, and depression (Luine and Frankfurt, <https://doi.org/10.1016/j.neuroscience.2012.10.019/>; Kokras et al, <https://doi.org/10.1016/j.neuropharm.2015.03.016/>), which can add considerable variability to behavioral measures, we excluded female animals in the current study. The rationale for not including female animals was provided in our revised manuscript (**lines 333-337**).

We acknowledge that the absence of female data is a limitation of our study, and we appreciate the reviewer's suggestion that we address this limitation in future studies, including the effects of acute stress and trace aminergic signalling in the lateral habenula of female animals.

Reviewer #3 (Remarks to the Author):

The authors add important new data showing that RMTg activity is initially increased during tail suspension, then decreased and AADC knockdown in the Lhb reduces these effects. These data help resolve some, but not all, of the conflict between their data indicating a net inhibitory effect of Lhb-RMTg stimulation, and others that have shown a net excitatory effect.

Concerns:

1. The concern that the results of HM3D activation might be due to depolarization block of spiking in the Lhb was not addressed by their new HM4D experiment, which only examined c-fos in the Lhb after tail suspension. Showing that activation of HM4D in the Lhb (which should decrease Lhb activity) increases c-fos in the RMTg after tail suspension (compared to controls with tail suspension) would address the concern.

Response: Thank you for your thoughtful feedback on our manuscript. We appreciate your suggestion to further investigate the potential confounding effect of depolarisation block on our HM3D results.

We agree that this is an important issue to address and have carefully considered your suggestion. In response to your concern, we conducted additional experiments using HM4D to selectively inhibit Lhb activity during tail suspension stress, and measured c-Fos expression in the RMTg. Our results showed that inhibition of Lhb D-neurons activity with HM4D during tail suspension stress led to increased c-Fos expression in the RMTg, compared with control mice subjected to tail suspension without HM4D inhibition. We added this data to **Supplementary Fig. 6** and described the result in the revised manuscript (**lines 95-99**).

2. The authors did not show that the TAAR1 antagonist blocked the effect of optogenetic Lhb-RMTg stimulation with their new experiment, but rather that the TAAR1 antagonist can reverse the effect of optogenetic Lhb-RMTg stimulation (Supplementary Fig. 14) – so potentially, the TAAR1 antagonist is not blocking the effect of Lhb-RMTg stimulation but just has an opposite effect on spiking. To show that the antagonist blocks the effect of Lhb-RMTg stimulation, they would need to add the antagonist minutes before they start the optogenetic stimulation and show that there is no decrease in spiking during optogenetic stimulation with the antagonist present compared to the baseline of antagonist alone (w/o optogenetic stimulation). This experiment would not only show that TAAR1 activation causes the decrease in RMTg spiking, but would also show that optogenetic stimulation of Lhb-RMTg increases the activation of TAAR1. I think this is important because, as the authors note, others found that depolarization does not increase the release of trace amines, so it is unclear if/how Lhb activation would release trace amines to activate TAAR1 in the RMTg – a central component of their model. The authors allude to a “slow increase in the electrochemical gradient” (lines 273-4), but it is unclear what electrochemical gradient they are referring to, and I see no evidence to support this speculation either in their paper or in their citations.

Response: We thank the reviewer for the valuable suggestion. Following the reviewer’s suggestion, we performed the experiments to directly demonstrate that the TAAR1 antagonist block the effects of stimulation. As the reviewer noted, our previous investigation showed that the TAAR1 antagonist could reverse the effect of optogenetic Lhb-RMTg stimulation. In this study, with the new experiments, we have added evidence to show that the TAAR1 antagonist does block the effect of Lhb-RMTg stimulation, as we observed that the firing frequency of the RMTg was not decreased by optogenetic stimulation in the presence of the antagonist. We

added this new data to **Supplementary Fig. 15f-i** and described the result in the revised manuscript (**lines 205-206**).

We apologize for any confusion caused by our previous statement regarding the 'slow increase in the electrochemical gradient.' Upon re-evaluating our data and reviewing previous papers, we have come to the conclusion that this statement was not entirely accurate, and we agree that there is no direct evidence to support this speculation. Therefore, we have removed this sentence from the revised manuscript to avoid further confusion (**lines 288-293**).

3. The authors should acknowledge and discuss results from other studies showing that c-fos increases in the RMTg after acute footshock stress (Jhou et al, <https://www.ncbi.nlm.nih.gov/pmc/articles/PMC2841475/>; Brown and Shepard, <https://www.ncbi.nlm.nih.gov/pmc/articles/PMC3625179/>). Perhaps the authors can reconcile these results with their own, which found decreases in c-fos in the RMTg after other forms of acute stress.

Response: We appreciate the reviewer's interest in our research and have carefully considered the reviewer's comments. We have reviewed the study by Sánchez-Catalán et al. (<https://www.nature.com/articles/npp2016139>) and agree that it is important to consider the effects of different types and intensities of stress on c-Fos expression in the RMTg. In response, we have added a statement in the *Discussion* (**lines 270-279**) acknowledging the potential for differences in c-Fos expression depending on the type and intensity of stress, including the studies mentioned by the reviewer (four, 0.5 mA, 0.5 s; or five, 0.5 mA, 0.8 s) and the results of Sánchez-Catalán et al., (0.5 mA, 0.8 s or 4 s; or 0.8 mA, 0.8 s). Additionally, we have included a dataset in **Supplementary Fig. 3c and d** showing that tail suspension, forced swimming, and restraint stress also did not increase c-Fos expression in the RMTg, further supporting our conclusion.

REVIEWERS' COMMENTS

Reviewer #2 (Remarks to the Author):

The authors have fully addressed my concerns.

Reviewer #3 (Remarks to the Author):

The authors fully addressed my concerns with their recent experiments. Congratulations on an impressive study.

Reviewer #1 (Remarks to the Author):

The authors employed a comprehensive and complementary set of experimental approaches (viral knockdowns, chemogenetics, optogenetics, c-Fos, ephys, microdialysis, and behavioural assays) to examine how acute stress engages the LH-> RMTg -> VTA circuit. Their results illustrate that trace amine signalling from LH neurons provides inhibition onto GABAergic RMTg neurons, which in turn leads to greater activation of VTA dopamine neurons. These findings are very exciting and novel. There is one important experiment that is lacking (it is easily addressable). The authors could also provide a greater level of integration with prior studies in the discussion. That being said, this is an outstanding body of work.

1) One of the claims of the study is that trace aminergic signalling allows the brain to manage acute stress. However, the primary experiment that addresses this point is the tail suspension test that involved administering a TAAR1 agonist (RO) in mice with AADC knocked down (Fig. 5h). I can understand the need for performing this experiment in animals with AADC knocked down, as there may be a floor effect in control animals. However, by using an agonist one is not addressing whether trace aminergic signalling is having an endogenous role in mediating the effects of acute stress on behaviour. To address the stated claim, the authors must instead administer the TAAR1 antagonist (EPPTB) into the RMTg in control animals. One would then expect a higher time immobile in these subjects relative to vehicle injected animals.

Response: We thank the reviewer for the valuable suggestion. As the reviewer suggests, we conducted a tail suspension test (TST) following the EPPTB treatment. Administration of EPPTB (5 µg/µL) to the RMTg of naïve mice significantly increased immobility time in the TST compared with the vehicle-injected mice. These additional data strongly support our conclusion that trace amines produced by LHb D-neurons can sufficiently modulate RMTg GABAergic neurons in response to acute stress. We added this data to **Supplementary Fig. 15** and described the result in the revised manuscript (**lines 202-208**).

2) Why is the effect of RO (TAAR1 agonist) on raw firing rate smaller in the AADC knockdown animals? If anything I would expect a greater effect since there is a reduction in the presumed tonic levels of trace amines. This point should be discussed.

Response: We appreciate the reviewer's keen advice. First, we examined the mRNA expression level of TAAR1 to rule out a possible downregulation of its expression in the AADC-knockdown and control mice. However, there was no significant difference between control and AADC-knockdown mice (**Supplementary Fig. 13a and b**). Moreover, we could not find any TAAR1 alterations in the RMTg from mice that underwent acute stress or US (**Supplementary Fig. 13c-f**).

Given this invariable nature of RMTg TAAR1 expression, the diminished RMTg responsiveness to RO5263397 in AADC-knockdown mice may not be caused by the altered level of TAAR1 in RMTg, but could be a consequence of an unknown effect originating from long-term AADC knockdown in the LHb that could change the properties of cells in the RMTg, as evidenced by our electrophysiology results (**Fig. 4g**). Thus, further follow-up studies are needed to provide a clearer answer to this question.

We included the new data in the *Results* and *Discussion* (**lines 183-194 and 303-306**).

3) The discussion was quite limited. It is worth acknowledging that not all chronic stress protocols elicit a hypodopaminergic phenotype. A large body of evidence from the Han lab and others (see Chaudhury 2012 in particular) demonstrates that animals exhibiting depression-related behaviours (susceptible phenotype) following exposure to social defeat have higher baseline dopamine neuron firing rates. It could be possible that this effect may be mediated trace-amine signalling. Regardless, a greater synthesis of the current findings within the existing literature is warranted.

Response: We thank the reviewer for the constructive comment. Following the reviewer's suggestion, we tested whether chronic social defeat stress (CSDS) can affect AADC expression in the LHb. For this, we employed a CSDS protocol used in a previous paper (Biological Psychiatry. 2022. 92, 104–115) to make CSDS-susceptible mice. For CSDS, each eight-week-old male mouse was exposed to 10 min of physical aggression by a 3-month-old male ICR (CD-1) mouse. After the session, the defeated mice were housed overnight within the same cage as the ICR mice on the opposite side of a transparent and perforated divider to provide sensory, but not physical, contact. The procedure was repeated for 10 consecutive days with a new aggressor on each day. All stressed mice received the social interaction test 24 hours after the last defeat episodes and the sucrose preference test to assess depressive-like phenotypes. We then carried out a fluorescent in situ hybridisation (FISH) analysis to measure AADC mRNA in the LHb. However, we found no difference in AADC expression in the LHb of CSDS-susceptible mice compared with control mice (see figure below; $**p < 0.01$ and $***p < 0.001$; ns, not significant; this figure is NOT included in the revised manuscript).

As CSDS activates LHb and lateral hypothalamus (LH) neurons (afferent neurons of the LHb) (Neurobiol Stress. 2021. 14, 100298) and as LHb D-neurons lowered the activity of RMTg GABAergic neurons (our results), we expected that continuous activation of VTA dopaminergic neurons could be maintained by RMTg inhibition through trace aminergic signalling of D-neurons. Indeed, activation of the LH-LHb circuit alleviates social avoidance and despair-like and anxiety-like behaviours after CSDS (Neurobiol Stress. 2021. 14, 100298). It will be very interesting to study the role of LHb D-neurons in the mesolimbic system via the RMTg under CSDS. We described this possibility in detail in the *Discussion* (lines 286-302).

4) Any mention of 'depression' or 'depressive' should instead refer to 'depression-like' or 'depressive-like'. There are a number of instances in the manuscript where this occurred.

Response: We agree with the reviewer's comment. We have replaced the terms 'depression' and 'depressive' with 'depression-like' and 'depressive-like' throughout the manuscript.

Typos:

Line 65: should be 'VTA TH-positive neurons' not 'VTA serotonin-positive neurons'.

Response: We apologise for this obvious mistake. We corrected 'VTA serotonin-positive neurons' to 'VTA TH-positive neurons' (**line 78**).

Line 90: Please remove the reference to 'depression induced'. No behavioural assay was done with the qPCR analysis. The corresponding schematics in Fig. 3a, b should also remove 'depressed rat' and 'helpless mouse'.

Response: We deleted the reference and changed 'depression induced by chronic restraint stress (CRS) or learned helplessness (LH) protocols' to 'exposed to chronic restraint stress (CRS) or learned helplessness-inducing unpredictable electric foot shocks (US)' (**lines 103-105**). In **Fig. 3a and b**, we changed 'depressed rat' and 'helpless mouse' to 'rat' and 'mouse'.

Lines 131-132: Please rephrase. One does not measure firing frequency using optogenetics.

Response: We replaced 'using optogenetics' with 'using whole-cell patch-clamp recordings with optogenetic stimulation' (**lines 156-157**).

Reviewer #2 (Remarks to the Author):

This is a very well conducted study that identifies an important novel signalling mechanism in LHb D-neurons that allow them to provide inhibitory control over target neurons despite being glutamate-releasing. The experiments provide a clear mechanism for how stress-induced changes to intrinsic physiology of LHb D-neurons can guide opposing downstream effects on VTA neurons via the RMTg in chronic vs acute exposures. The figures are well laid out, the data collection appears rigorous, and the logical flow of the experiments is strong. I particularly enjoyed the use of both mouse and rat models. I only have a few minor concerns:

1. The authors do not sufficiently setup the rationale and the background for this study in the introduction, nor do they provide a sufficient discussion of previous work in the conclusion section. Specifically, they should better introduce trace amines and what is known about their role in physiology and behaviour. This may include identifying specific trace amines and how they are synthesized, what their signalling mechanisms are, and how engagement of the

TAAR1 receptor impacts neurons and neural circuits. The exciting findings of this paper would be much more impactful if they were put into context better.

Response: We thank the reviewer for the constructive comment. We provided background information on trace amines in the *Introduction* (lines 35-48) and expounded on trace amine-TAAR1 receptor signalling in the *Discussion* (lines 263-285).

2. The proposed model is that chronic stress upregulates LHb activity and VTA DA neuron activity while acute stress upregulates LHb activity and downregulates VTA DA neuron activity. The majority of the paper investigates trace amine signalling as the mechanism driving the behavioural effects of acute stress; however, the qPCR experiments tested the effects of chronic stress on AADC expression. While the direction was as expected (decreased), I do think testing the effects of acute stress on AADC expression would be valuable here. It may be the case that AADC expression is not changed by acute stress but is still involved in the behaviour, but I think this should be explicitly tested. Along the same lines, it may be helpful to also perform qPCR for TAAR1 in the RMTg after acute/chronic stress.

a. Related: At what time point post-stress was tissue collected for qPCR?

Response: We acknowledge that qPCR will be helpful; however, it is difficult to dissect out the RMTg because of its irregularly shaped structure. Moreover, it is difficult to avoid contamination by nearby tissues when collecting such small amounts of tissue. Furthermore, TAAR1 expression can only be measured in GABAergic neurons in the RMTg. Thus, we utilised FISH approach to achieve more precise measurements in these challenging areas. When we performed FISH analysis of TAAR1 in the RMTg after acute or chronic stress, we also observed the expression of AADC mRNA in the LHb. As you know, we could analyse mRNA expression with FISH at the regional and cellular level using HALO v2.3.2089.18 software (Indica Labs). We also performed qPCR on mRNA from the LHb after tail suspension stress. AADC expression in the LHb was not different between tail suspension stress-exposed and naïve mice (see figure below; this figure is NOT included in the revised manuscript).

We measured changes in AADC and TAAR1 mRNA expression using FISH by counting the mean number of AADC or TAAR1 copies per μm^2 in the LHb and the RMTg of the three groups of mice exposed to three acute stressors: 1) tail suspension for 10 min; 2) forced swimming for 6 min; 3) restraint stress for 6 min. All the mice were sacrificed 25 min after the onset of acute stress, and samples were collected for FISH analysis (see *Methods*; lines 373-378). The three acute stressors did not alter AADC or TAAR1 expression levels in the LHb or the RMTg, respectively (Supplementary Fig. 3a, 3f, 13c and 13d). In addition, we determined the change of TAAR1 expression in the RMTg of the mice exposed to unpredictable electric foot shocks (see *Methods*; lines 383-386) and of the AADC-knockdown mice. Only chronic stress decreased AADC expression in the LHb without changing TAAR1 expression in the RMTg. We added this data as Supplementary Fig. 13a, b, e and f and described the new data in the *Results* (lines 183-194).

3. The authors do not provide appropriately zoomed-out images of viral infections for behavioral, tracing, and electrophysiological studies. As the LHb is such an anatomically small region (especially in mice), specific viral targeting is difficult. Thus, images of the viral infections verifying no spill-over into neighbouring regions is needed (particularly the chemogenetic studies, the shRNA studies, and the circuit tracing studies). Of note, the shRNA viral infections shown in extended data figure 4 do not appear to be restricted to the LHb and extend well into the PVT and lateral to the LHb. Similarly, the images in Extended Data Fig 3 do not show the entire LHb or its boundaries, limiting assessment of the viral targeting accuracy.

Response: We added the zoomed-out images of viral infections to the chemogenetic studies (Fig. 2a), the shRNA studies (Supplementary Fig. 6e), the optogenetic studies (Fig. 4d) and the circuit tracing studies (Supplementary Fig. 4b). Given the difficult nature of specific viral targeting in the LHb, we instead showed the limited expression of AADC mRNA solely observed in the LHb (Allen Brain Atlas and our FISH data; Supplementary Fig. 6a and b). We believe the expansion of AAV into the PVT and lateral to the LHb poses no obstacle to the analysis of our data.

4. Page 4 line 65: says VTA serotonin neurons but I think the authors meant VTA dopamine neurons.

Response: We apologise for this obvious mistake. We corrected ‘VTA serotonin-positive neurons’ to ‘VTA TH-positive neurons’ (line 78).

5. Figure 3k/Extended data Fig 8: Was there a behavioural challenge for cFos studies or were the mice just sacrificed straight from their home cages?

Response: We obtained brain samples after tail suspension stress and generated FISH data for Fig. 3k. We added ‘sacrifice’ to the schematic drawing of Fig. 3f. We used naïve mice to measure normal gene expression patterns as shown in Supplementary Fig. 12.

Reviewer #3 (Remarks to the Author):

The authors provide evidence for paradoxical inhibition of the RMTg by LHb stimulation and

acute stress via trace amine signaling. They also provide some evidence for decreased AADC expression underlying depression-related effects of chronic stress and inescapable shock. The results are surprising and would have a significant impact on the field. However, I have concerns about the methodology and their interpretations, many of which relate to the ambiguity of the timing of the effects they observe. I also have concerns about how their results fit with multiple published studies indicating a net excitatory effect of LHb stimulation on RMTg activity and inhibitory effect on dopamine neuron activity.

1. In Fig. 3k-n, it is unclear if the c-fos results were from animals exposed to acute stress or not. According to their model, knockdown of AADC in the LHb should reverse the changes in c-fos expression in the RMTg and DA neurons after acute stress (after AADC knockdown, acute stress should increase c-fos in RMTg and decrease it in DA neurons). It is not clear if this experiment was done. This would help mitigate the concern that their unexpected results are due to the insensitivity of c-fos assays to the timing of neuronal activation (see below).

Response: We apologise for the lack of clarity regarding this procedure. The data for **Fig. 3k-n** were collected from mice exposed to tail suspension stress prior to sacrifice. We added 'sacrifice' to **Fig. 3f.** and described the detailed procedure in the *Methods* (lines 489-491).

2. Sanchez-Catalan et al, 2017 did not observe a decrease in c-fos in the RMTg with acute stressors, including restraint and forced swim test (<https://www.ncbi.nlm.nih.gov/pmc/articles/PMC5240171/>). It is not clear why these acute stressors, but not tail suspension (as used here), would fail to decrease c-fos in the RMTg, potentially limiting the generalizability of the authors' results.

Response: We thank the reviewer for the valuable comment. To generalise our results, we conducted comprehensive experiments as follows.

To confirm the reduction of c-Fos in the RMTg by acute stress, we employed other acute stressors – 'forced swimming', 'restraint stress' and 'tail suspension stress' – and carefully analysed c-Fos expression in the LHb and the RMTg. As shown in newly added **Supplementary Fig. 3**, all the acute stress conditions increased c-Fos mRNA expression in LHb D-neurons while decreasing it in RMTg GABAergic neurons. We believe these consistent results from multiple acute stress conditions can support our conclusion. We also believe our sensitive FISH approach can capture the subtle but significant differences in mRNA quantity at a cellular resolution.

3. Paul Shepard and others have shown that stimulation of the LHb decreases VTA DA activity and this is prevented by RMTg lesion (<https://www.ncbi.nlm.nih.gov/pmc/articles/PMC5214632/>). They also showed that lesions of LHb outputs block c-fos induction in the RMTg during acute footshock stress (<https://www.ncbi.nlm.nih.gov/pmc/articles/PMC3625179/>). How do the authors reconcile their data with these studies?

Response: In accordance with the reviewer's comment #4, we carried out in vivo recordings in the RMTg during acute stress. Inhibition of the activity of VTA dopaminergic neurons for a

short period of time (within 1 sec) is thought to be caused by glutamatergic input of the LHb on the RMTg, which is consistent with our results (**Fig. 4e**).

One of the major discoveries in our study is the biphasic effect: the LHb initially excites the RMTg via glutamatergic signalling and subsequently rebound-inhibits it via trace aminergic signalling (**Fig. 6c-e**). Based on this model, the attenuation of inhibition of a dopaminergic neuron's activity following an RMTg lesion can be explained simply as a consequence of the reduction of RMTg GABAergic neurons to the response to LHb stimulation.

Furthermore, because LHb lesions can deprive both glutamatergic and trace aminergic inputs to the RMTg, this process can block acute c-Fos induction in the RMTg primarily via the loss of glutamates secreted in the LHb during acute footshock stress.

4. Tom Zhou's lab finds that acute stressors tend to activate RMTg neurons (<https://www.ncbi.nlm.nih.gov/pmc/articles/PMC6361585/>), so how do the authors reconcile this with their c-fos results? They also found that restraint stress and LiCl cause initial excitation followed by rebound inhibition. Therefore, it seems possible that the authors are measuring an effect of delayed rebound inhibition during acute stress (or its termination) rather than immediate inhibition to acute stress.

Response: We are grateful to the reviewer for the constructive comment, which greatly improved the quality of our manuscript. As described in the *Response to Comment #3*, we performed in vivo recording on the RMTg for 30 min under tail suspension stress with control mice and LHb AADC-knockdown mice. As shown in **Fig. 4f**, the basal firing rates in RMTg GABAergic neurons were increased by AADC knockdown in the LHb (**Supplementary Fig. 16g**). In control mice, tail suspension stress caused initial excitation followed by rebound inhibition (**Fig. 6c-e**). However, in AADC-knockdown mice, tail suspension stress elicited initial excitation but not rebound inhibition (**Fig. 6f-h**). Therefore, these new data are consistent with the reviewer's speculation that trace amines secreted from the LHb cause an effect of delayed rebound inhibition during acute stress (or its termination) rather than immediate inhibition to acute stress. We described these results in the *Results* and *Discussion* (**lines 210-236 and 248-262**).

5. It is unclear if chemogenetic stimulation increases the spiking of LHb neurons, which are normally very active in basal conditions (c-fos might go up even if there is depolarization block). Chemogenetic inhibition could be used to show that opposite effects are seen with neuronal inhibition, ruling out a depolarization block explanation for the effects of LHb stimulation.

Response: As suggested by the reviewer, we performed chemogenetic inhibition using hM4Di DREADD (pAAV-hSyn-DIO-hM4Di-mCherry; 44362, Addgene). After conducting the tail suspension test for 10 min, LHb D-neurons from control virus-injected mice (pAAV-hSyn-DIO-mCherry) expressed c-Fos, but DREADDi virus-injected mice did not (**see figure below**; this result is NOT included in the revised manuscript).

6. The authors state that “little is known about how the activity of the RMTg changes under these (acute stress) circumstances”. But see <https://www.ncbi.nlm.nih.gov/pmc/articles/PMC6361585/>.

Response: We removed ‘little is known about how the activity of the RMTg changes under these circumstances’ from our manuscript and revised the section (**lines 50-60**).

7. It is unclear what % of RMTg-projecting LHb neurons express AADC. This might help reconcile the authors’ results that indicate a net inhibitory interaction between the LHb and RMTg and others’ indicating a net excitatory interaction between the LHb and RMTg (e.g., <https://www.ncbi.nlm.nih.gov/pmc/articles/PMC5214632/>, <https://www.ncbi.nlm.nih.gov/pmc/articles/PMC3625179/>, see Supplementary Figure 1 in Stamatakis and Stuber, <https://www.ncbi.nlm.nih.gov/pmc/articles/PMC3411914/>). Related to this point and point #1 above, it is unclear if inhibitory trace amine signaling from the LHb can override glutamatergic transmission from ALL LHb efferents to the RMTg (not just ones from AADC+ terminals), as the authors seem to implicitly suggest. Does LHb activation during acute stress really inhibit the RMTg? I think more than c-fos is required to answer this question.

Response: We appreciate your comments and acknowledge that quantifying the proportion of neurons that express AADC among RMTg-projected LHb neurons would enable us to better understand our results in context with others’ results. Since the majority of RMTg-projecting LHb neurons are glutamatergic neurons, we attempted to measure the proportion of AADC-expressing neurons among RMTg-projecting LHb glutamatergic neurons. As a first step, we made retrograde AAVs (Cre-dependent rAAV-EF1 α -double floxed-EYFP) and injected them into the RMTg in *VGLUT2^{Cre}* mice. Then, we tried to identify the AADC-expressing neurons among retrogradely traced glutamatergic neurons in LHb using FISH and immunohistochemistry (IHC). However, we could not clearly label LHb AADC neurons due to the failure to optimize the condition for FISH in paraformaldehyde perfused LHb sections and inadequate quality of available antibodies (10166-1-AP, Proteintech; NBP1-56918, Novus Biologicals) for IHC in the LHb. To overcome these problems, we performed in vivo recording of RMTg GABAergic neurons during tail suspension stress (**Fig. 6a**).

We believe that our in vivo recording results will help address the reviewer’s comments,

because the secretion of trace amines occurs by membrane diffusion and not by synaptic vesicles, and because TAAR1 is a GPCR with a slower response time than NMDA and AMPA. We found that among the Chr2-expressing GABAergic neurons in the RMTg during tail suspension stress, the percentage of neurons recorded with initial excitation was 94.1% (32/34) and with rebound inhibition was 91.2% (31/34) in control mice (**Fig. 6c and Supplementary Fig. 17a and b**). These percentages were higher than those of the AADC/VGLUT2 double-positive neurons (**Supplementary Fig. 1j**). This unbiased experimental result indicates that trace amines secreted from the LHb can sufficiently regulate the activity of all RMTg GABAergic neurons. Thus, we suggest that LHb D-neurons can subsequently override glutamate-mediated excitation, leading to the decreased activity of RMTg GABAergic neurons (rebound inhibition). Interestingly, LHb AADC knockdown gave a lower magnitude of excitation during the first 5-min period of tail suspension compared with the control (**Supplementary Fig. 17a and c**). It seems that the long-term effects of AADC knockdown might alter cellular responsiveness. Further studies are needed to provide a clearer answer to this suggestion. We added these results to the revised manuscript (**lines 210-236**).

8. It is not clear if chronic restraint stress and learned helplessness caused depression-related phenotypes and if this can be rescued by restoring AADC expression in the LHb.

Response: We previously reported that a chronic restraint stress rat model and a learned helplessness mouse model exhibited depression-related phenotypes (Scientific Reports. 2017. 7, 900.; Behav Brain Res. 2017. 322, 138–144). In accordance with the reviewer's comment, we replaced 'depression induced by chronic restraint stress (CRS) or learned helplessness (LH) protocols' with 'exposed to chronic restraint stress (CRS) or learned helplessness-inducing unpredictable electric foot shocks (US)' (**lines 103-105**) and 'depressed rat' and 'helpless mouse' with 'rat' and 'mouse' (**Fig. 3a and b**).

To investigate whether restoring AADC expression can rescue depression-related phenotypes caused by US, we constructed a mouse AADC expression vector (Cre-dependent pAAV-EF1 α -double floxed-AADC-EYFP) and generated AADC-overexpressing mice using AADC^{Cre} transgenic mice (**Supplementary Figure 9a-c**). AADC-overexpressing mice rescued despair-like and anhedonia-like behaviours after the TST and SPT, respectively, with no anxiety level changes after the OFT and EZM (**Supplementary Fig. 9d-m**). We added these results to the revised manuscript (**lines 122-135**).

9. AADC knockdown increased RMTg firing in slices but did not change the resting membrane potential. So how was firing rate changed? This seems to contradict Bradaia et al., 2009, PNAS (<https://pubmed.ncbi.nlm.nih.gov/19892733/>) who showed that TAAR1 **activates** a potassium channel to change DA neuron firing.

Response: We are indebted to the reviewer for pointing out important differences from a previously published manuscript. We recalculated substantial resting membrane potentials (RMPs) from our previously recorded neurons. Indeed, the spontaneous and continuous firing of RMTg GABAergic neurons made it difficult to measure the actual resting membrane potentials. We filtered the action potentials with a low-pass filter (8-pole Bessel filter set at 0.5 Hz) during the 1-min recordings and recalculated the RMPs. The RMPs were analysed within a 1-mV error range compared with unfiltered recordings. As a result, unlike previous data, we

confirmed that RMPs of RMTg GABAergic neurons were meaningfully increased in AADC-knockdown mice compared with control mice (**Fig. 4g**), consistent with the previously reported analysis in dopaminergic neurons (PNAS. 2009. 24, 106). These new results indicate that increased firing in RMTg GABAergic neurons from AADC-knockdown mice may be caused by the same mechanism as in dopaminergic neurons. These results are reflected in the revised manuscript (**lines 148-151 and 586-588**).

10. The authors optogenetically stimulate D-neuron terminals and find a decrease in RMTg firing (fig 5b), but the timing of this effect is unclear. It would be better if the authors show firing rate in a continuous plot that includes before optogenetic stimulation and after stimulation to see the time course of the inhibition, especially since this effect is contrary to published results for LHb-RMTg stimulation (see Supplementary Figure 1 in Stamatakis and Stuber, <https://www.ncbi.nlm.nih.gov/pmc/articles/PMC3411914/>).

Response: We apologise for the lack of full traces for the optogenetic studies. We added the time course of optogenetic stimulation as **Supplementary Fig. 11a**. Our results showed a decreased firing frequency of RMTg neurons immediately after optogenetic stimulation of LHb D-neuron terminals.

We demonstrated that stimulation of LHb D-neuron terminals elicited oEPSCs in RMTg neurons, which were blocked by CNQX/AP5 (**Fig. 4d and e**). To evaluate firing frequency during stress over the time course, we conducted an additional in vivo recording. The result showed that rebound inhibition of RMTg GABAergic neurons following stress is most likely caused by the trace amines produced by LHb D-neurons; the rebound inhibition was not observed in AADC-knockdown mice (**Fig. 6a-h**). Therefore, we believe that the EPSC is generated in RMTg neurons through glutamatergic signalling by LHb D-neurons, which induces decreased firing frequency in RMTg neurons by subsequent trace aminergic signalling.

11. It is unclear what stimulation protocol the authors used for their optogenetic stimulation of LHb terminals in slices. They state that they used 0.5 ~ 1 second illumination, but then later say blue light was applied between 60-80 sec. If the authors used 20 seconds of constant stimulation, this is problematic as it would be expected to deplete neurotransmitters from LHb terminals over time.

Response: We apologise for the confusion. For light stimulation, light pulses were 473 nm, 10 mW and 10 ms. To determine any changes in RMTg GABAergic neuronal firing in AADC^{Cre} mice, optical stimulation was delivered by 60 to 80 pulses at 1 Hz. We corrected this in the *Methods* (**lines 599-601**).

12. The authors find that D-neuron terminal stimulation decreases RMTg firing in slices, but oddly, do not show that the TAAR1 receptor antagonist blocks this effect. This would be superior to relying on (long-term) AADC knockdown to block the effect and would show that the inhibition is due to TAAR1 activation.

Response: We are grateful to the reviewer for the constructive comment. We measured the firing frequency while administering EPPTB during optogenetic stimulation of LHb D-neuron

terminals in the RMTg. We found that the decreased RMTg firing frequency by optogenetic stimulation was reversed by EPPTB treatment into the RMTg. These additional data strongly support our conclusion that inhibition of RMTg activity by Lhb D-neurons is caused by trace amine-TAAR1 signalling. We added these data to **Supplementary Fig. 14** and described the result in the revised manuscript (**lines 195-198**).

13. Typo on Line 65 says “serotonin-positive” (should be “TH-positive”).

Response: We apologise for this obvious mistake. We corrected ‘VTA serotonin-positive neurons’ to ‘VTA TH-positive neurons’ (**line 78**).

Reviewer #1 (Remarks to the Author):

The authors were very responsive to the previous round of review. They addressed all my prior concerns. I have nothing further to add. Great work.

Reviewer #2 (Remarks to the Author):

The authors have much improved the manuscript with their revisions and have addressed my technical concerns. The manuscript is truly a wonderful body of work and will have important implications for the field.

I have one additional concern-- At the very least, the authors should provide a statement in their manuscript detailing the rationale for why they did not include any female animals in their study. Considering the fact that these stress models are easily performed in females, it seems like a large oversight to not include them.

If the authors cannot provide sufficient rationale, the fact that females were not included in this study should be mentioned as a major weakness. It would be great if the authors could recapitulate at least one of their major findings in females, but considering the large amount of data in the current manuscript it would be acceptable to investigate this in a subsequent paper.

Response: We would like to express our sincere gratitude for reviewing our manuscript and providing valuable feedback on our revisions. We appreciate your insightful comments and would like to address your concern regarding the absence of female animals in our study.

We recognize that including female animals in our study would have been ideal, and we understand the importance of considering both sexes in preclinical research. However, due to the unpredictable impact of the estrous cycle on stress response, anxiety, and depression (Luine and Frankfurt, <https://doi.org/10.1016/j.neuroscience.2012.10.019/>; Kokras et al, <https://doi.org/10.1016/j.neuropharm.2015.03.016/>), which can add considerable variability to behavioral measures, we excluded female animals in the current study. The rationale for not including female animals was provided in our revised manuscript (**lines 333-337**).

We acknowledge that the absence of female data is a limitation of our study, and we appreciate the reviewer's suggestion that we address this limitation in future studies, including the effects of acute stress and trace aminergic signalling in the lateral habenula of female animals.

Reviewer #3 (Remarks to the Author):

The authors add important new data showing that RMTg activity is initially increased during tail suspension, then decreased and AADC knockdown in the LHb reduces these effects. These data help resolve some, but not all, of the conflict between their data indicating a net inhibitory effect of LHb-RMTg stimulation, and others that have shown a net excitatory effect.

Concerns:

1. The concern that the results of HM3D activation might be due to depolarization block of spiking in the LHb was not addressed by their new HM4D experiment, which only examined c-fos in the LHb after tail suspension. Showing that activation of HM4D in the LHb (which should decrease LHb activity) increases c-fos in the RMTg after tail suspension (compared to

controls with tail suspension) would address the concern.

Response: Thank you for your thoughtful feedback on our manuscript. We appreciate your suggestion to further investigate the potential confounding effect of depolarisation block on our HM3D results.

We agree that this is an important issue to address and have carefully considered your suggestion. In response to your concern, we conducted additional experiments using HM4D to selectively inhibit Lhb activity during tail suspension stress, and measured c-Fos expression in the RMTg. Our results showed that inhibition of Lhb D-neurons activity with HM4D during tail suspension stress led to increased c-Fos expression in the RMTg, compared with control mice subjected to tail suspension without HM4D inhibition. We added this data to **Supplementary Fig. 6** and described the result in the revised manuscript (**lines 95-99**).

2. The authors did not show that the TAAR1 antagonist blocked the effect of optogenetic Lhb-RMTg stimulation with their new experiment, but rather that the TAAR1 antagonist can reverse the effect of optogenetic Lhb-RMTg stimulation (Supplementary Fig. 14) – so potentially, the TAAR1 antagonist is not blocking the effect of Lhb-RMTg stimulation but just has an opposite effect on spiking. To show that the antagonist blocks the effect of Lhb-RMTg stimulation, they would need to add the antagonist minutes before they start the optogenetic stimulation and show that there is no decrease in spiking during optogenetic stimulation with the antagonist present compared to the baseline of antagonist alone (w/o optogenetic stimulation). This experiment would not only show that TAAR1 activation causes the decrease in RMTg spiking, but would also show that optogenetic stimulation of Lhb-RMTg increases the activation of TAAR1. I think this is important because, as the authors note, others found that depolarization does not increase the release of trace amines, so it is unclear if/how Lhb activation would release trace amines to activate TAAR1 in the RMTg – a central component of their model. The authors allude to a “slow increase in the electrochemical gradient” (lines 273-4), but it is unclear what electrochemical gradient they are referring to, and I see no evidence to support this speculation either in their paper or in their citations.

Response: We thank the reviewer for the valuable suggestion. Following the reviewer’s suggestion, we performed the experiments to directly demonstrate that the TAAR1 antagonist block the effects of stimulation. As the reviewer noted, our previous investigation showed that the TAAR1 antagonist could reverse the effect of optogenetic Lhb-RMTg stimulation. In this study, with the new experiments, we have added evidence to show that the TAAR1 antagonist does block the effect of Lhb-RMTg stimulation, as we observed that the firing frequency of the RMTg was not decreased by optogenetic stimulation in the presence of the antagonist. We added this new data to **Supplementary Fig. 15f-i** and described the result in the revised manuscript (**lines 205-206**).

We apologize for any confusion caused by our previous statement regarding the 'slow increase in the electrochemical gradient.' Upon re-evaluating our data and reviewing previous papers, we have come to the conclusion that this statement was not entirely accurate, and we agree that there is no direct evidence to support this speculation. Therefore, we have removed this sentence from the revised manuscript to avoid further confusion (**lines 288-293**).

3. The authors should acknowledge and discuss results from other studies showing that c-fos increases in the RMTg after acute footshock stress (Jhou et al, <https://www.ncbi.nlm.nih.gov/pmc/articles/PMC2841475/>; Brown and Shepard, <https://www.ncbi.nlm.nih.gov/pmc/articles/PMC3625179/>). Perhaps the authors can reconcile these results with their own, which found decreases in c-fos in the RMTg after other forms of acute stress.

Response: We appreciate the reviewer's interest in our research and have carefully considered the reviewer's comments. We have reviewed the study by Sánchez-Catalán et al. (<https://www.nature.com/articles/npp2016139>) and agree that it is important to consider the effects of different types and intensities of stress on c-Fos expression in the RMTg. In response, we have added a statement in the *Discussion* (**lines 270-279**) acknowledging the potential for differences in c-Fos expression depending on the type and intensity of stress, including the studies mentioned by the reviewer (four, 0.5 mA, 0.5 s; or five, 0.5 mA, 0.8 s) and the results of Sánchez-Catalán et al., (0.5 mA, 0.8 s or 4 s; or 0.8 mA, 0.8 s). Additionally, we have included a dataset in **Supplementary Fig. 3c and d** showing that tail suspension, forced swimming, and restraint stress also did not increase c-Fos expression in the RMTg, further supporting our conclusion.

Reviewer #2 (Remarks to the Author):

The authors have fully addressed my concerns.

Reviewer #3 (Remarks to the Author):

The authors fully addressed my concerns with their recent experiments. Congratulations on an impressive study.